# Model Immunization from a Condition Number Perspective

**Amber Yijia Zheng** [* 1]  **Cedar Site Bai** [* 1]  **Brian Bullins** [1]  **Raymond A. Yeh** [1]

## Abstract

Model immunization aims to pre-train models that are difficult to fine-tune on harmful tasks while retaining their utility on other non-harmful tasks. Though prior work has shown empirical evidence for immunizing text-to-image models, the key understanding of when immunization is possible and a precise definition of an immunized model remain unclear. In this work, we propose a framework, based on the condition number of a Hessian matrix, to analyze model immunization for linear models. Building on this framework, we design an algorithm with regularization terms to control the resulting condition numbers after pre-training. Empirical results on linear models and non-linear deep-nets demonstrate the effectiveness of the proposed algorithm on model immunization. The code is available at https://github.com/amberyzheng/model-immunization-cond-num.

## 1. Introduction

Model immunization, recently proposed by Zheng & Yeh (2024), studies how to pre-train a model that is more difficult to fine-tune on harmful content, but not others. The aim is to mitigate the risk of misuse (Brundage et al., 2018; Marchal et al., 2024) associated with open-sourced models by immunizing them before they are released to the public.

Zheng & Yeh (2024) focus on immunizing text-to-image models, where they formulate immunization as a bi-level optimization. Empirically, they show that pre-trained diffusion models that undergo immunization are more difficult to fine-tune on a given harmful concept dataset. To quantify this difficulty, they compare the generation quality of models with and without immunization after a fixed number of fine-tuning iterations. While the empirical results are promising,

a definition of an immunized model and the circumstances that make immunization possible remain unclear.

To tackle this issue, we propose a framework to study model immunization using the condition number (Gloub & Van Loan, 1996). The effectiveness of immunization can be characterized by the condition number of the Hessian matrix. When using gradient-based methods during fine-tuning, a condition number closer to one indicates faster convergence (Boyd & Vandenberghe, 2004), *i.e.*, easier to fine-tune. With this perspective, we observe that the existence of an effective immunization for linear models is related to the angle between the singular vectors of the harmful fine-tuning dataset's covariance matrix and the pre-training dataset's covariance matrix.

From this condition number perspective, we propose an immunization algorithm to find such a model. In detail, we propose two additional terms to regularize the condition number during pre-training. Each of the introduced regularization terms can be shown to ensure a monotonic increase/decrease of the condition number under gradient updates.

Beyond the theoretical results, we empirically validate the proposed algorithm on linear models for regression and image classification tasks. Lastly, we conduct experiments using the proposed algorithm on non-linear models, *i.e.*, deep-nets. Despite the gap in theory, we observe that the proposed approach remains effective at model immunization across ResNet (He et al., 2016) and ViT (Dosovitskiy, 2021).

**Our contributions are summarized as follows:**

- We introduce a framework based on the condition number to study the task of model immunization. This framework leads to a concrete definition of an immunized model along with a novel experiment setup and evaluation metric to compare the quality of different immunization techniques.
- We propose regularizers to maximize/minimize the condition number, with a guaranteed monotonic increase/decrease when updated with the gradient-based method.
- Together with the task objective and regularizers, we demonstrate that the proposed algorithm effectively immunizes linear models and deep-nets on regression/image classification tasks.

---

[*]Equal contribution  [1]Department of Computer Science, Purdue University. Correspondence to: Raymond A. Yeh <rayyeh@purdue.edu>.

*Proceedings of the 42$^{nd}$ International Conference on Machine Learning*, Vancouver, Canada. PMLR 267, 2025. Copyright 2025 by the author(s).

## 2. Preliminaries

This section provides the background of the condition number and its connection to gradient descent. Additionally, we briefly review transfer learning (Zhuang et al., 2020), as it can be a technique for misusing open-source models.

**Condition number and convergence of gradient descent.** Given a general matrix $\boldsymbol{S}$, the condition number (Gloub & Van Loan, 1996) is defined as

$$\kappa(\boldsymbol{S}) \triangleq \|\boldsymbol{S}\|_2 \left\|\boldsymbol{S}^\dagger\right\|_2 = \sigma_{\boldsymbol{S}}^{\texttt{max}}/\sigma_{\boldsymbol{S}}^{\texttt{min}}, \qquad (1)$$

where $\dagger$ is the pseudoinverse and $\sigma_{\boldsymbol{S}}$ corresponds to the max/min singular value of $\boldsymbol{S}$. The condition number is related to the convergence rate of gradient-based algorithms.

Consider an optimization problem $\min_{\mathbf{w}} \mathcal{L}(\mathbf{w})$ where $\mathcal{L}$ is strongly convex and has a Hessian $\nabla^2 \mathcal{L}$ with max/min singular values denoted as $\sigma^{\texttt{max/min}}$. In this case, the constant step-size steepest descent algorithm has a convergence rate (Bubeck, 2015) of the following

$$\|\mathbf{w}_t - \mathbf{w}^*\|^2 \leq \left(1 - \frac{\sigma^{\texttt{min}}}{\sigma^{\texttt{max}}}\right)^t \|\mathbf{w}_0 - \mathbf{w}^*\|^2, \qquad (2)$$

where $\mathbf{w}^*$ denotes the optimal solution, and $\mathbf{w}^t$ denotes the steepest descent iterate at step $t$. We can observe that a larger condition number corresponds to a slower convergence.

**Condition number regularization.** Nenov et al. (2024) proposed a regularizer for minimizing the condition number of some general matrix $\boldsymbol{S}$

$$\mathcal{R}_{\texttt{well}}(\boldsymbol{S}) = \frac{1}{2}\|\boldsymbol{S}\|_2^2 - \frac{1}{2p}\|\boldsymbol{S}\|_F^2, \qquad (3)$$

in which $p$ is the minimum dimension of $\boldsymbol{S}$, and the norms correspond to the spectral norm and Frobenius norm. They showed that $\mathcal{R}_{\texttt{well}}(\boldsymbol{S})$ is a valid regularizer by proving its nonnegativity, and is an upper bound on $\log(\kappa(\boldsymbol{S}))$. In addition, they showed that $\mathcal{R}_{\texttt{well}}(\boldsymbol{S})$ is differentiable under some mild conditions, and if updated with gradient descent, it is guaranteed to decrease the condition number monotonically. See Appendix A for the exact statements.

Different from Nenov et al. (2024), we propose a differentiable regularizer that is guaranteed to **increase** the condition number as an upper bound on $1/\log(\kappa(\boldsymbol{S}))$. For model immunization, instead of a general matrix $\boldsymbol{S}$, we need to consider the regularization of the Hessian of linear models composed of a feature extractor and a classifier, while preserving their differentiability and monotonicity guarantees during gradient updates to the feature extractor.

**Transfer learning via linear probing.** In this work, we focus on the transfer learning method of linear probing. Given a pre-trained feature extractor $f_\theta : \mathbb{R}^{D_{\texttt{in}}} \to \mathbb{R}^{D_{\texttt{hid}}}$, linear probing learns an a linear classifier $h_{\mathbf{w}} : \mathbb{R}^{D_{\texttt{hid}}} \to$

$\mathbb{R}^{D_{\texttt{out}}}$ over the target dataset $\mathcal{D} = \{(\boldsymbol{x}, \boldsymbol{y})\}$ using the frozen feature extractor $f_\theta$. This model learning is formulated as the following optimization problem

$$\min_{\mathbf{w}} \mathcal{L}(\mathcal{D}, \mathbf{w}, \theta) \triangleq \min_{\mathbf{w}} \sum_{(\boldsymbol{x}, \boldsymbol{y}) \in \mathcal{D}} \ell(h_{\mathbf{w}} \circ f_\theta(\boldsymbol{x}), \boldsymbol{y}) \qquad (4)$$

where $\ell$ denotes a suitable loss function, *e.g.*, cross-entropy. By keeping $\theta$ fixed, the model leverages features learned from pre-training task and transfers them to the target task. This approach is effective when the target dataset is too small to train a model from scratch.

## 3. Immunization with Condition Number

The goal of model immunization is to learn a pre-trained model $g_\omega \circ f_{\theta^{\texttt{I}}}$, consisting of a classifier $g_\omega$ and an immunized feature extractor $f_{\theta^{\texttt{I}}}$, such that fine-tuning $f_{\theta^{\texttt{I}}}$ on a harmful task is difficult, but not for other tasks. The model should also maintain a good pre-training task performance. Specifically, we study the setting when a bad actor uses linear probing on a pre-trained linear feature extractor with gradient descent.

**Immunization setting.** We denote a pre-training dataset as $\mathcal{D}_{\texttt{P}} = \{(\mathbf{x}, \mathbf{y})\}$ and a harmful dataset as $\mathcal{D}_{\texttt{H}} = \{(\mathbf{x}, \tilde{\mathbf{y}})\}$ where $\mathbf{x} \in \mathbb{R}^{D_{\texttt{in}}}$. The bad actor performs linear probing using $\mathcal{D}_{\texttt{H}}$ following Eq. (4) with an $\ell_2$ loss. We will focus our analysis on linear pre-trained feature extractor without dimensionality reduction, *i.e.*, $f_\theta \triangleq \mathbf{x}^\top \theta$ with $\theta \in \mathbb{R}^{D_{\texttt{in}} \times D_{\texttt{in}}}$.

**Definition 3.1.** Under this setting, a model is said to be *immunized* if it satisfies the following:

**(a)** It is more difficult to apply linear probing on the harmful task $\mathcal{D}_{\texttt{H}}$ using the immunized feature extractor $f_{\theta^{\texttt{I}}}$ than directly on the input data, *i.e.*,

$$\kappa(\nabla_{\mathbf{w}}^2 \mathcal{L}(\mathcal{D}_{\texttt{H}}, \mathbf{w}, \theta^{\texttt{I}})) \gg \kappa(\nabla_{\mathbf{w}}^2 \mathcal{L}(\mathcal{D}_{\texttt{H}}, \mathbf{w}, \boldsymbol{I})), \qquad (5)$$

where $\boldsymbol{I}$ denotes the identity matrix.

**(b)** It is not more difficult to apply linear probing on other tasks. As there is only one other task $\mathcal{D}_{\texttt{P}}$, an immunized feature extractor should have

$$\kappa(\nabla_{\omega}^2 \mathcal{L}(\mathcal{D}_{\texttt{P}}, \omega, \theta^{\texttt{I}})) \leq \kappa(\nabla_{\omega}^2 \mathcal{L}(\mathcal{D}_{\texttt{P}}, \omega, \boldsymbol{I})). \qquad (6)$$

Note: we use $\omega$ to denote the classifier parameters of the pre-training task and $\mathbf{w}$ for the harmful task.

**(c)** The immunized model should maintain a competitive task performance on the pre-training dataset $\mathcal{D}_{\texttt{P}}$, *i.e.*,

$$\min_{\omega, \theta} \mathcal{L}(\mathcal{D}_{\texttt{P}}, \omega, \theta) \approx \min_{\omega} \mathcal{L}(\mathcal{D}_{\texttt{P}}, \omega, \theta^{\texttt{I}}). \qquad (7)$$

For linear models, as long as $\theta^{\texttt{I}}$ is invertible, exact equality can be achieved.

## 3.1. Analysis on Immunized Linear Models

To provide some intuition on how the feature extractor $\theta$ affects the convergence of linear probing, we study the analytical form of the singular values of the Hessian. For readability, we will rewrite linear probing in Eq. (4) by considering $f_\theta \triangleq \mathbf{x}^\top \theta$ and a $\ell_2$-loss.

Let $\boldsymbol{X}_{\text{H}} \in \mathbb{R}^{N \times D_{\text{in}}}$ and $\boldsymbol{Y}_{\text{H}} \in \mathbb{R}^{N \times D_{\text{out}}}$ denote data from $\mathcal{D}_H$ stacked into matrices with $N \triangleq |\mathcal{D}_{\text{H}}|$. When using a $\ell_2$-loss, Eq. (4) can be written as

$$\min_{\mathbf{w}} \mathcal{L}(\mathcal{D}_{\text{H}}, \mathbf{w}, \theta) = \min_{\mathbf{w}} \|(\boldsymbol{X}_{\text{H}}\theta)\mathbf{w} - \boldsymbol{Y}\|_2^2. \quad (8)$$

In this case, the Hessian matrix

$$\boldsymbol{H}_{\text{H}}(\theta) \triangleq \nabla_{\mathbf{w}}^2 \mathcal{L}(\mathcal{D}_{\text{H}}, \mathbf{w}, \theta) = \theta^\top \boldsymbol{K}_{\text{H}}\theta, \quad (9)$$

where $\boldsymbol{K}_{\text{H}} \triangleq \boldsymbol{X}_{\text{H}}^\top \boldsymbol{X}_{\text{H}}$ is the data covariance matrix.

**Proposition 3.2.** *The singular values of the Hessian matrix in Eq.* (9) *are given by*

$$\sigma_i = \sum_{j=1}^{D_{\text{in}}} \left(\sigma_{\theta,i}(\boldsymbol{u}_{\theta,i}^\top \boldsymbol{q}_j)\sqrt{\gamma_j}\right)^2, \quad \forall i \in \{1, \dots, D^{\text{in}}\}. \quad (10)$$

*Here, $\sigma_{\theta,i}$ and $\boldsymbol{u}_{\theta,i}$ correspond to the $i$-th singular value and vector of $\theta$. Next, $\gamma_j$ and $\boldsymbol{q}_j$ correspond to the $j$-th singular value and vector of the covariance $\boldsymbol{K}$.*

*Proof sketch.* This result can be shown by using the fact that $\boldsymbol{K}_{\text{H}}$ is a symmetric positive semi-definite matrix and decomposing via SVD. The complete proof is provided in Appendix B.1. □

From Eq. (10), we can see that the singular value of the Hessian depends on the relative angle between the singular vectors between feature extractor $\theta$ and the covariance matrix of the data $\boldsymbol{K}_{\text{H}}$. As the feature extractor is shared between the pretrained $\mathcal{D}_{\text{P}}$ and harmful $\mathcal{D}_{\text{H}}$ datasets, the strength of the immunization depends on the relative angle between the singular vectors of $\boldsymbol{K}_{\text{P}}$ and $\boldsymbol{K}_{\text{H}}$. For example, if the singular vectors (sorted by the singular values) are all perfectly aligned between the two, then no $\theta$ can simultaneously maximize $\kappa(\nabla_{\mathbf{w}}^2 \mathcal{L}(\mathcal{D}_{\text{H}}, \mathbf{w}, \theta))$ and minimize $\kappa(\nabla_\omega^2 \mathcal{L}(\mathcal{D}_{\text{P}}, \omega, \theta))$.

With a better understanding of the effect of the feature extractor $\theta$ on the condition number, we will next present an algorithm to immunize a model.

## 4. Algorithm for Immunizing a Model

We formulate model immunization as an optimization problem with the following objective:

$$\min_{\omega, \theta} \mathcal{R}_{\text{ill}}(\boldsymbol{H}_{\text{H}}(\theta)) + \mathcal{R}_{\text{well}}(\boldsymbol{H}_{\text{P}}(\theta)) + \mathcal{L}(\mathcal{D}_{\text{P}}, \omega, \theta), \quad (11)$$

where $\mathcal{R}_{\text{ill}}$, to be defined in Sec. 4.1, denotes our proposed regularizer to maximize the condition number, $\mathcal{R}_{\text{well}}$

**Algorithm 1** Condition number regularized gradient descent for model immunization

**input** Primary task $\mathcal{D}_{\text{P}} = (\boldsymbol{X}_{\text{P}}, \boldsymbol{Y}_{\text{P}})$, harmful task input $\boldsymbol{X}_{\text{H}}$, supervised loss $\mathcal{L}$, learning rate $\eta$, regularizing constants $\lambda_{\text{P}}, \lambda_{\text{H}} \in \mathbb{R}_+$, model initialization $\theta_0, \omega_0$
1: $\boldsymbol{K}_{\text{P}} = \boldsymbol{X}_{\text{P}}^\top \boldsymbol{X}_{\text{P}}$
2: $\boldsymbol{K}_{\text{H}} = \boldsymbol{X}_{\text{H}}^\top \boldsymbol{X}_{\text{H}}$
3: **for** $t = 0, 1, \dots, T-1$ **do**
4: $\quad \omega_{t+1} = \omega_t - \eta\nabla_\omega \mathcal{L}(\omega_t, \theta_t; \mathcal{D}_{\text{P}})$
5: $\quad \boldsymbol{H}_{\text{P}}(\theta_t) = \theta_t^\top \boldsymbol{K}_{\text{P}}\theta_t, \; \boldsymbol{H}_{\text{H}}(\theta_t) = \theta_t^\top \boldsymbol{K}_{\text{H}}\theta_t$
6: $\quad \theta_{t+1} = \theta_t - \eta\nabla_\theta \mathcal{L}(\omega_t, \theta_t; \boldsymbol{X}_1)$
$$\quad - \eta\lambda_{\text{P}}\boldsymbol{K}_P^{-1}\nabla_\theta \mathcal{R}_{\text{well}}(\boldsymbol{H}_{\text{P}}(\theta_t))$$
$$\quad - \eta\lambda_{\text{H}}\boldsymbol{K}_H^{-1}\nabla_\theta \mathcal{R}_{\text{ill}}(\boldsymbol{H}_{\text{H}}(\theta_t))$$
7: **end for**
**output** Immunized feature extractor $\theta_{\text{I}} \triangleq \theta_T$.

in Eq. (3) denotes the regularizer to minimize the condition number, $\boldsymbol{H}_{\text{P}}(\theta) \triangleq \nabla_\omega^2 \mathcal{L}(\mathcal{D}_{\text{P}}, \omega, \theta) = \theta^\top \boldsymbol{K}_{\text{P}}\theta$ is the Hessian matrix of the pre-training task, and $\mathcal{L}$ denotes the supervised loss.

Each of the terms encourages the model to satisfy the three immunization requirements in Definition 3.1. For readability, we have dropped the scalar hyperparameters balancing the terms. We propose to solve Eq. (11) using a gradient-based method as outlined in Alg. 1.

In the remainder of this section, we will first introduce the novel regularizer to *maximize* general matrices' condition number and their relevant properties (Sec. 4.1). We then show how to incorporate the regularizers $\mathcal{R}_{\text{ill}}$ and $\mathcal{R}_{\text{well}}$ into the immunization setup (Sec. 4.2). Finally, we discuss the provable guarantees with respect to each of the regularizers (Sec. 4.3).

### 4.1. Regularizer for Maximizing the Condition Number

We analyze the condition number of a general matrix $\boldsymbol{S} \in \mathbb{R}^{p_r \times p_c}$, $p = \min\{p_r, p_c\}$, and $\text{rank}(\boldsymbol{S}) = k \leq p$. The compact SVD of $\boldsymbol{S}$ is given by $\boldsymbol{S} = \boldsymbol{U}\text{Diag}(\boldsymbol{\sigma})\boldsymbol{V}^\top$, in which $\boldsymbol{\sigma} = [\sigma_1, \cdots, \sigma_k]^\top$ such that $\sigma_{\boldsymbol{S}}^{\text{max}} = \sigma_1 \geq \sigma_2 \geq \cdots \geq \sigma_k = \sigma_{\boldsymbol{S}}^{\text{min}} > 0$ and $\boldsymbol{u}_i, \boldsymbol{v}_i$ denotes the $i^{th}$ column vector of $\boldsymbol{U}, \boldsymbol{V}$ for $i \in [k]$.

Inspired by the regularizer for minimizing the condition number, we propose its counterpart for maximizing the condition number

$$\mathcal{R}_{\text{ill}}(\boldsymbol{S}) = \frac{1}{\frac{1}{2k}\|\boldsymbol{S}\|_F^2 - \frac{1}{2}(\sigma_{\boldsymbol{S}}^{\text{min}})^2}, \quad (12)$$

which satisfies the properties in the following theorem.

**Theorem 4.1** (Properties of $\kappa$-maximizing regularizer $\mathcal{R}_{\text{ill}}(\boldsymbol{S})$)**.**

*(1) [Nonnegativity]* For any $\boldsymbol{S} \in \mathbb{R}^{p_r \times p_c}$, $\mathcal{R}_{\mathtt{ill}}\left(\boldsymbol{S}\right) \geq 0$, and $\mathcal{R}_{\mathtt{ill}}\left(\boldsymbol{S}\right) = 0$ if and only if $\kappa\left(\boldsymbol{S}\right) = \infty$.

*(2) [Upper Bound]* $\frac{1}{\log(\kappa(\boldsymbol{S}))} \leq \left(\sigma_{\boldsymbol{S}}^{\mathtt{max}}\right)^2 \mathcal{R}_{\mathtt{ill}}\left(\boldsymbol{S}\right)$, i.e., $\mathcal{R}_{\mathtt{ill}}(\boldsymbol{S})$ upper bounds $\frac{1}{\log(\kappa(\boldsymbol{S}))}$ when $\sigma_{\boldsymbol{S}}^{\mathtt{max}}$ is reasonably away from $\infty$.

*(3) [Differentiability]* If $\sigma_{\boldsymbol{S}}^{\mathtt{min}} = \sigma_k < \sigma_i$ for any $i < k$, i.e., $\sigma_{\boldsymbol{S}}^{\mathtt{min}}$ is unique, then $\mathcal{R}_{\mathtt{ill}}\left(\boldsymbol{S}\right)$ is differentiable and

$$\nabla_{\boldsymbol{S}} \mathcal{R}_{\mathtt{ill}}(\boldsymbol{S}) = \frac{\sigma_k \boldsymbol{u}_k \boldsymbol{v}_k^\top - \frac{1}{k}\boldsymbol{S}}{\left(\frac{1}{2k}\|\boldsymbol{S}\|_F^2 - \frac{1}{2}\left(\sigma_{\boldsymbol{S}}^{\mathtt{min}}\right)^2\right)^2}. \quad (13)$$

*(4) [Monotonic Increase]* If $\sigma_{\boldsymbol{S}}^{\mathtt{min}}$ is unique, update $\boldsymbol{S}$ with $\nabla_{\boldsymbol{S}} \mathcal{R}_{\mathtt{ill}}(\boldsymbol{S})$ such that $\boldsymbol{S}' = \boldsymbol{S} - \eta_2 \nabla_{\boldsymbol{S}} \mathcal{R}_{\mathtt{ill}}(\boldsymbol{S})$ for $0 < \eta_2 < \frac{k}{k-1}\left(\frac{1}{2k}\|\boldsymbol{S}\|_F^2 - \frac{1}{2}\left(\sigma_{\boldsymbol{S}}^{\mathtt{min}}\right)^2\right)^2$, then $\kappa\left(\boldsymbol{S}'\right) > \kappa\left(\boldsymbol{S}\right)$.

*Proof sketch.* We provide some intuitive illustrations of the proof and defer the complete version to Appendix B.2.

For (1), as the squared Frobenius norm of a matrix equals the sum of the squares of its singular values, the denominator of $\mathcal{R}_{\mathtt{ill}}\left(\boldsymbol{S}\right)$ is the average of the squared singular values minus their minimum, ensuring it is nonnegative. It can be shown that $\mathcal{R}_{\mathtt{ill}}\left(\boldsymbol{S}\right)$ is inversely related to $\kappa\left(\boldsymbol{S}\right)$, which indicates that $\mathcal{R}_{\mathtt{ill}}\left(\boldsymbol{S}\right) = 0$ if and only if $\kappa\left(\boldsymbol{S}\right) = \infty$.

For (2) the upper bound holds by the design of $\mathcal{R}_{\mathtt{ill}}\left(\boldsymbol{S}\right)$ and applying the mean value inequality on

$$\log\left(\kappa(\boldsymbol{S})^2\right) = \log\left(\left(\sigma_{\boldsymbol{S}}^{\mathtt{max}}\right)^2\right) - \log\left(\left(\sigma_{\boldsymbol{S}}^{\mathtt{min}}\right)^2\right). \quad (14)$$

For (3), even though $\sigma_{\boldsymbol{S}}^{\mathtt{min}}$ is not differentiable since it involves taking the minimum of the singular values, its subdifferential is well-defined (Lewis, 1995). When $\sigma_{\boldsymbol{S}}^{\mathtt{min}}$ is unique, its subdifferential reduces to a singleton, *i.e.*, its gradient, making $\mathcal{R}_{\mathtt{ill}}\left(\boldsymbol{S}\right)$ also differentiable.

For (4), one key observation is that the closed-form $\nabla_{\boldsymbol{S}} \mathcal{R}_{\mathtt{ill}}(\boldsymbol{S})$ shares the same set of singular vectors as $\boldsymbol{S}$, so that the linear relation in gradient update can be passed on to singular values. By choosing a suitable step size, the increase in condition number can be guaranteed. $\square$

Theorem 4.1 demonstrates that the regularizer $\mathcal{R}_{\mathtt{ill}}\left(\boldsymbol{S}\right)$ introduced is a reasonable upper bound for maximizing condition numbers and indicates that under some mild condition, *i.e.*, the minimum singular value is unique, simple first-order algorithms like gradient descent can be used to minimize the regularizer with guaranteed increase in condition number.

## 4.2. Incorporating Regularizers into Immunization

Given the immunization setup, we now analyze the regularizer $\mathcal{R}_{\mathtt{ill}}$ and $\mathcal{R}_{\mathtt{well}}$ for matrices with the specific structure of feature covariance matrices, and propose the corresponding algorithm for model immunization.

As illustrated in the immunization setup, the feature extractor $\theta$ is the trainable parameter. For data $\boldsymbol{X} \in \mathbb{R}^{N \times D_{\mathtt{in}}}$ of the feature extractor, we analyze the condition number of $\boldsymbol{H}(\theta) \triangleq \theta^\top \boldsymbol{K}\theta \in \mathbb{R}^{D_{\mathtt{in}} \times D_{\mathtt{in}}}$ with $\mathrm{rank}\left(\boldsymbol{H}\right) = k$, and compact SVD $\boldsymbol{H} = \boldsymbol{U}\mathrm{Diag}(\boldsymbol{\sigma})\boldsymbol{V}^\top$. Recall, we define $\boldsymbol{K} = \boldsymbol{X}^\top \boldsymbol{X}$ to be the covariance matrix of the data.

In the following theorem, we show that under the same conditions, the introduced regularizers $\mathcal{R}_{\mathtt{ill}}\left(\cdot\right)$ and $\mathcal{R}_{\mathtt{well}}\left(\cdot\right)$ are also differentiable w.r.t. $\theta$ when applied to $\theta^\top \boldsymbol{K}\theta$.

**Theorem 4.2.** *For $\boldsymbol{H}\left(\theta\right) = \theta^\top \boldsymbol{K}\theta$, if its maximum and minimum singular values $\sigma_1$ and $\sigma_k$ are unique, then*

*(1)* $\nabla_\theta \mathcal{R}_{\mathtt{well}}\left(\boldsymbol{H}\left(\theta\right)\right) = 2\boldsymbol{K}\theta\left(\sigma_1 \boldsymbol{v}_1 \boldsymbol{v}_1^\top - \frac{1}{D_{\mathtt{in}}}\theta^\top \boldsymbol{K}\theta\right)$,

*(2)* $\nabla_\theta \mathcal{R}_{\mathtt{ill}}\left(\boldsymbol{H}\left(\theta\right)\right) = \frac{2\boldsymbol{K}\theta\left(\sigma_k \boldsymbol{v}_k \boldsymbol{v}_k^\top - \frac{1}{k}\theta^\top \boldsymbol{K}\theta\right)}{\left(\frac{1}{2k}\|\theta^\top \boldsymbol{K}\theta\|_F^2 - \frac{1}{2}\sigma_k^2\right)^2}$.

*Proof sketch.* The differentiability follows from the same argument of Theorem 4.1 (3) under the condition that the maximum and minimum singular values are unique. The closed-form gradients are computed with the chain rule in matrix calculus defined by the Frobenius inner product. The complete proof can be found in Appendix B.3. $\square$

With the closed-form gradient of the regularizers w.r.t. $\theta$, we propose our algorithm for model immunization in Alg. 1. Specifically, Alg. 1 employs the general gradient descent framework. Line 4 conducts standard updates for the classifier $\omega$, minimizing the supervised loss $\mathcal{L}$. In lines 5 to 6, the regularizers $\mathcal{R}_{\mathtt{ill}}$ and $\mathcal{R}_{\mathtt{well}}$ are applied on the feature covariance $\boldsymbol{H}_{\mathtt{H}}(\theta)$ of the harmful task and $\boldsymbol{H}_{\mathtt{P}}(\theta)$ of the pretraining task. This is done by updating the feature extractor $\theta$ with the gradients $\nabla_\theta \mathcal{R}_{\mathtt{ill}}(\boldsymbol{H}_{\mathtt{H}})$ and $\nabla_\theta \mathcal{R}_{\mathtt{well}}(\boldsymbol{H}_{\mathtt{P}})$ normalized by their input covariances and the gradient from the supervised loss $\nabla_\theta \mathcal{L}$.

## 4.3. Condition Number Guarantees

We show in the following theorem that the condition number decrease/increase guarantees introduced in Theorem A.1 (4) and Theorem 4.1 (4) are preserved for $\theta^\top \boldsymbol{K}\theta$ even when the gradient updates are taken in $\theta$ as in Alg. 1, instead of $\theta^\top \boldsymbol{K}\theta$.

**Theorem 4.3.** *For the trainable feature extractor $\theta$, feature covariance $\boldsymbol{H}_{\mathtt{P}}\left(\theta\right) = \theta^\top \boldsymbol{K}_{\mathtt{P}}\theta$ of the primary task and $\boldsymbol{H}_{\mathtt{H}}\left(\theta\right) = \theta^\top \boldsymbol{K}_{\mathtt{H}}\theta$ of the immunization task with $\mathrm{rank}\left(\boldsymbol{H}_{\mathtt{P}}\right) = k_{\mathtt{P}}$, $\mathrm{rank}\left(\boldsymbol{H}_{\mathtt{H}}\right) = k_{\mathtt{H}}$ and compact SVD $\boldsymbol{H}_{\mathtt{P}}\left(\theta\right) = \boldsymbol{U}_{\mathtt{P}}\mathrm{Diag}(\boldsymbol{\sigma}_{\mathtt{P}})\boldsymbol{V}_{\mathtt{P}}^\top$, $\boldsymbol{H}_{\mathtt{H}}\left(\theta\right) = \boldsymbol{U}_{\mathtt{H}}\mathrm{Diag}(\boldsymbol{\sigma}_{\mathtt{H}})\boldsymbol{V}_{\mathtt{H}}^\top$, for $\boldsymbol{\sigma}_{\mathtt{P}} = [\sigma_{\mathtt{P},1}, \cdots, \sigma_{\mathtt{P},k_{\mathtt{P}}}]$, $\boldsymbol{\sigma}_{\mathtt{H}} = [\sigma_{\mathtt{H},1}, \cdots, \sigma_{\mathtt{H},k_{\mathtt{H}}}]$,*

*(1)* *if $\sigma_{\boldsymbol{H}_{\mathtt{P}}}^{\mathtt{max}}$ is unique, i.e., $\sigma_{\boldsymbol{H}_{\mathtt{P}}}^{\mathtt{max}} = \sigma_{\mathtt{P},1} > \sigma_{\mathtt{P},2}$, update $\theta$ such that $\theta' = \theta - \eta_{\mathtt{P}}\boldsymbol{K}_{\mathtt{P}}^{-1}\nabla_\theta \mathcal{R}_{\mathtt{well}}(\boldsymbol{H}_{\mathtt{P}}\left(\theta\right))$ for*

$$0 < \eta_{\mathtt{P}} < \min\left\{\frac{1}{\left(1 - \frac{1}{D_{\mathtt{in}}}\right)\sigma_{\mathtt{P},1}}, \frac{\sqrt{\sigma_{\mathtt{P},1}\sigma_{\mathtt{P},2}} - \sigma_{\mathtt{P},2}}{\frac{2}{D_{\mathtt{in}}}\sigma_{\mathtt{P},2}^2}\right\}, \textit{ then}$$

$$\kappa\left(\theta'^{\top}\boldsymbol{K}_{\mathtt{P}}\theta'\right) < \kappa\left(\theta^{\top}\boldsymbol{K}_{\mathtt{P}}\theta\right),$$

(2) *if $\sigma_{\boldsymbol{H}_{\mathtt{H}}}^{\mathtt{min}}$ is unique,* i.e., $\sigma_{\boldsymbol{H}_{\mathtt{H}}}^{\mathtt{min}} = \sigma_{\mathtt{H},k_{\mathtt{H}}} < \sigma_{\mathtt{H},k_{\mathtt{H}}-1}$, *update $\theta$ such that* $\theta' = \theta - \eta_{\mathtt{H}}\boldsymbol{K}_{\mathtt{H}}^{-1}\nabla_{\theta}\mathcal{R}_{\mathtt{ill}}(\boldsymbol{H}_{\mathtt{H}}(\theta))$ *for* $0 < \eta_{\mathtt{H}} < \frac{1}{1-2\sigma_{\boldsymbol{H}_{\mathtt{H}}}^{\mathtt{min}}/k_{\mathtt{H}}}\left(\frac{1}{2k_{\mathtt{H}}}\left\|\theta^{\top}\boldsymbol{K}_{\mathtt{H}}\theta\right\|_{F}^{2} - \frac{1}{2}\left(\sigma_{\boldsymbol{H}_{\mathtt{H}}}^{\mathtt{min}}\right)^{2}\right)^{2}$, *then* $\kappa\left(\theta'^{\top}\boldsymbol{K}_{\mathtt{H}}\theta'\right) > \kappa\left(\theta^{\top}\boldsymbol{K}_{\mathtt{H}}\theta\right)$.

*Proof sketch.* There is a mismatch between the gradient update on $\theta$ and the condition number update, which is observed for $\boldsymbol{H}(\theta)$. To address this, we carefully leverage the structure of the problem, noting that $\boldsymbol{H}(\theta)$, unlike a general matrix, is symmetric and positive semidefinite, with identical left and right singular vectors. Exploiting this property, along with our algorithm design, ensures that the linearity in singular value updates is preserved when expanding $\boldsymbol{H}(\theta')$ using the closed-form gradient in Theorem 4.2. Consequently, a monotonic increase or decrease in the condition number can be guaranteed by appropriately selecting the step size. The full proof is provided in Appendix B.4. □

### 4.4. Additional Discussion

**Implementation considerations.** At a glance, it may seem that to implement Alg. 1 using automatic differentiation packages, *e.g.*, Pytorch (Paszke et al., 2019), one would have to implement a custom optimizer and involve multiple update steps. Instead, we observe that by directly modifying the computation graph, it would only involve a single backward pass. This is done by introducing a "dummy layer" with an identified function as its forward pass and its backward pass multiplies the gradient by the inverse feature covariance matrix. The "dummy layer" implementation is inspired by prior works in gradient estimator (Bengio et al., 2013; Roeder et al., 2017). Pseudo-code is provided in Appendix C.3.

**Limitations.** The monotonicity guarantees in Theorem 4.3 serve as a theoretical justification for our proposed algorithm, albeit a partial reflection of the application setup. Note that the feature extractor is updated with the gradients of the two regularizers jointly together with that of the supervised loss and the guarantees may not linearly combine as such. In practice, maintaining the balance between $\kappa\left(\boldsymbol{H}_{\mathtt{P}}(\theta)\right)$ and $\kappa\left(\boldsymbol{H}_{\mathtt{H}}(\theta)\right)$ requires a proper choice of hyperparameters.

Next, the current framework we analyzed focuses on linear feature extractors and using linear probing for transfer learning. We are aware of the practical limitations of this setting. To address this, in the experiments, we empirically study the effect of the proposed method on non-linear models, *i.e.*, deep-nets, and demonstrate our method's potential despite the theoretical gap.

*Table 1.* Quantitative results of immunization in House Price dataset (Montoya & DataCanary, 2016), computed over 5 random seeds.

| Method | Eq. (15) (i)↑ | Eq. (15) (ii)↓ | RIR↑ |
|---|---|---|---|
| $\mathcal{R}_{\mathtt{ill}}$ *Only* | **90.02** ±**3.773** | 72.415 ±3.545 | 1.244 ±0.021 |
| *IMMA* | 7.053 ±1.662 | 3.545 ±0.880 | 2.001 ±0.187 |
| *Opt $\kappa$* | 1.518 ±0.027 | **0.016** ±**0.001** | 92.58 ±4.492 |
| **Ours** | 18.92 ±2.056 | 0.053 ±0.002 | **356.20** ±**5.491** |

## 5. Experiments

We evaluate the proposed Alg. 1 on regression and image classification tasks using linear models, and also explored immunizing non-linear models, *i.e.*, deep-nets. Experiment and implementation details are provided in Appendix C.

**Evaluation metrics.** We introduce the *relative immunization ratio (RIR)* to quantify the effectiveness of the immunization based on the ratio of the condition number of Hessian, defined as follows:

$$\mathtt{RIR} \triangleq \underbrace{\left(\frac{\kappa(\boldsymbol{H}_{\mathtt{H}}(\theta_{\mathtt{I}}))}{\kappa(\boldsymbol{H}_{\mathtt{H}}(\boldsymbol{I}))}\right)}_{\text{(i)}} \Big/ \underbrace{\left(\frac{\kappa(\boldsymbol{H}_{\mathtt{P}}(\theta_{\mathtt{I}}))}{\kappa(\boldsymbol{H}_{\mathtt{P}}(\boldsymbol{I}))}\right)}_{\text{(ii)}} \quad (15)$$

where $\boldsymbol{I}$ denotes the identity matrix. Each term here measures the ratio between condition numbers with and without the pre-trained feature extractor on the (i) harmful task or (ii) on the pre-training task.

A successful immunization is characterized by:

(i) a large ratio $\frac{\kappa(\boldsymbol{H}_{\mathtt{H}}(\theta_{\mathtt{I}}))}{\kappa(\boldsymbol{H}_{\mathtt{H}}(\boldsymbol{I}))}$, *i.e.*, using the immunized feature extractor makes the optimization of linear probing more difficult on the harmful task.

(ii) a small ratio $\frac{\kappa(\boldsymbol{H}_{\mathtt{P}}(\theta_{\mathtt{I}}))}{\kappa(\boldsymbol{H}_{\mathtt{P}}(\boldsymbol{I}))}$), *i.e.*, using the pre-trained extractor *do not* make optimization more difficult on the pre-training task.

To obtain a single metric, we compare (i) and (ii) relative to each other. In other words, an effective immunized model should have a relative immunization ratio $\mathtt{RIR} \gg 1$.

**Baselines.** We consider three baselines for comparisons:

- $\mathcal{R}_{\mathtt{ill}}$ *Only* immunizes the model by minimizing only the regularizer $\mathcal{R}_{\mathtt{ill}}(\boldsymbol{H}_{\mathtt{H}})$ as defined in Eq. (12) using gradient descent.
- *IMMA* (Zheng & Yeh, 2024) is formulated as a bi-level optimization program where both lower and upper tasks are solved via gradient descent. In the lower-level, it minimizes $\mathcal{L}(\mathcal{D}_{\mathtt{H}}, \mathbf{w}, \theta)$ w.r.t. $\theta$ to obtain $\theta^{\star}$, and in the upper-level, it maximizes $\mathcal{L}(\mathcal{D}_{\mathtt{H}}, \mathbf{w}, \theta^{\star}) - \mathcal{L}(\mathcal{D}_{\mathtt{P}}, \omega, \theta^{\star})$ w.r.t. $\theta$ by backpropagating through $\theta^{\star}$.
- *Opt $\kappa$* directly minimizes $\kappa(\boldsymbol{H}_{\mathtt{P}}(\theta)) - \kappa(\boldsymbol{H}_{\mathtt{H}}(\theta))$ w.r.t. $\theta$

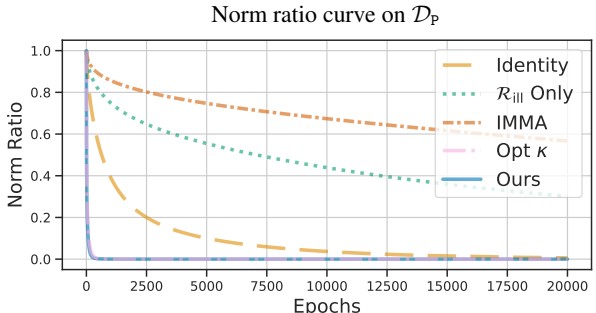
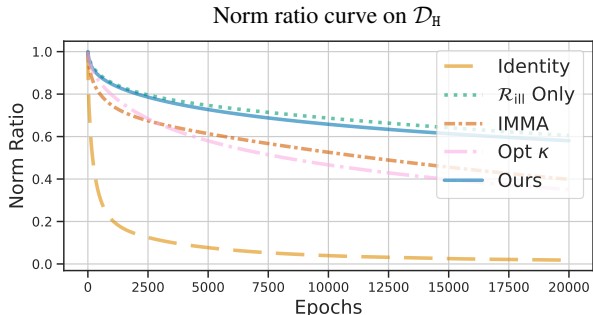

*Figure 1.* Norm ratio Eq. (16) *vs.* Epochs. We visualize the convergence of linear probing of different immunized models using gradient descent with an exact line search. Here, *Identity* corresponds to not using a feature extractor, *i.e.*, $\theta_{\mathtt{I}} = \boldsymbol{I}$. Observe that Ours made the convergence faster on $\mathcal{D}_{\mathtt{P}}$ while slower in $\mathcal{D}_{\mathtt{H}}$ when compared to the other baselines; consistent with the results in Tab. 1.

via gradient descent instead of using our proposed regularizers.

### 5.1. Experiments on Immunizing Linear Models

**Linear regression task.** We use the regression task from the House prices dataset (Montoya & DataCanary, 2016). We split the data into $\mathcal{D}_{\mathtt{P}}$ and $\mathcal{D}_{\mathtt{H}}$ based on the feature `MSZoning`. For the pre-training task, we use the target of `LotArea` and for the harmful task we use the target of `SalePrice`. Both $\mathcal{D}_{\mathtt{P}}$ and $\mathcal{D}_{\mathtt{H}}$ contain input vectors of dimension 79. We immunized the model by running Alg. 1 for 100 epochs with $\eta = 0.005$. We choose $\lambda_{\mathtt{P}}$ and $\lambda_{\mathtt{H}}$ by balancing the gradient norm of $\mathcal{R}_{\mathtt{well}}$ and $\mathcal{R}_{\mathtt{ill}}$. The implementation details can be found in Appendix C.2.

In Tab. 1, we present the empirical results of immunizing a linear feature extractor $\theta$. We observe that only *Opt $\kappa$* and our method successfully immunize the model achieving an RIR that's much greater than 1. For $\mathcal{R}_{\mathtt{ill}}$ *Only* and *IMMA*, while they successfully made the harmful task more ill-conditioned, *i.e.*, *Eq.* (15) (i) went up, however, this is at the cost of making the other task ill-conditioned as well, *i.e.*, *Eq.* (15) (ii) went up.

Next, we demonstrate how a large condition number slows down the convergence of linear probing on the harmful task by analyzing the norm ratio defined as

$$\|\mathbf{w}_t - \mathbf{w}^\star\|_2^2 / \|\mathbf{w}_0 - \mathbf{w}^\star\|_2^2, \tag{16}$$

which measures how the classifier weights $\mathbf{w}_t$ at step $t$ approach the optimal weights $\mathbf{w}^\star$ during fine-tuning. Note, naively choosing a step size will not reflect the difference in condition number. Hence, we use the exact line search (Boyd & Vandenberghe, 2004) which chooses the step size that minimizes the loss at each iteration.

As illustrated in Fig. 1, both our method and *Opt $\kappa$* slow down convergence in $\mathcal{D}_{\mathtt{H}}$ compared to Identity while accelerating convergence in $\mathcal{D}_{\mathtt{P}}$. Furthermore, our method

*Table 2.* Quantitative results of immunization in MNIST (LeCun, 1998), computed over 3 random seeds and averaged over all digit pairs. Note that *Opt $\kappa$* has large STD in RIR, resulting in the deviation between RIR and the ratio of the averaged values.

| Method | Eq. (15) (i)↑ | Eq. (15) (ii) ↓ | RIR ↑ |
|---|---|---|---|
| $\mathcal{R}_{\mathtt{ill}}$ *Only* | **14.832** ±1.039 | 8.654 ±0.606 | 1.933 ±0.046 |
| *IMMA* | 4.522 ±0.139 | 2.774 ±0.094 | 1.774 ±0.041 |
| *Opt $\kappa$* | 3.196 ±1.225 | 0.756 ±1.171 | 69.73 ±54.00 |
| **Ours** | 6.345 ±0.188 | **0.149** ±0.009 | **70.04** ±3.280 |

achieves a stronger immunization effect than *Opt $\kappa$*. In contrast, $\mathcal{R}_{\mathtt{ill}}$ *Only* and *IMMA* slowed the convergence on both the harmful task $\mathcal{D}_{\mathtt{H}}$ and the pre-training task $\mathcal{D}_{\mathtt{P}}$.

**Image classification task.** For image classification, we conduct experiments using MNIST (LeCun, 1998). The MNIST dataset consists of images over 10-digit classes, which can be formulated into 10 independent binary classification tasks. Across all pairs of tasks, we choose one to be the harmful task $\mathcal{D}_{\mathtt{H}}$ and the other the pre-training $\mathcal{D}_{\mathtt{P}}$ resulting in a total of 90 experiments. We ran Alg. 1 for 30 epochs with $\eta = 0.005$ for these experiments. The implementation details can be found in Appendix C.2.

In Tab. 2, we present the quantitative results on these binary task pairs. For each entry, the values are averaged over all 90 pairs. Based on the averaged results, we observe that our method effectively immunizes the linear feature extractor $\theta$ on $\mathcal{D}_{\mathtt{H}}$ without compromising performance on $\mathcal{D}_{\mathtt{P}}$. Although *Opt $\kappa$* achieves comparable RIR with our method, the variances of the metric values are relatively large. This indicates that *Opt $\kappa$* is sensitive to random initialization while our method is robust.

In Fig. 2 we further analyze the results by visualizing the $\log(\mathtt{RIR})$ for each digit pair. A blue block indicates successful immunization, while a red block indicates failure. It can

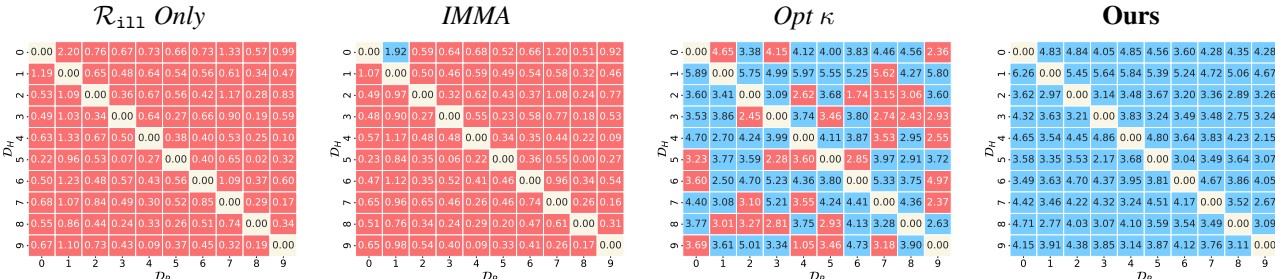

*Figure 2.* Visualization of log(**RIR**) **of binary classification tasks created from MNIST.** Each element in the figure corresponds to the log(RIR) of a model immunized against $\mathcal{D}_\text{H}$ from the pre-training task of $\mathcal{D}_\text{P}$. We color the block blue if RIR $\gg 1$, and red otherwise. Our method succeeds in immunizing the model across all digit pairs, while the baselines failed in most pairs.

be observed that $\mathcal{R}_\text{ill}$ *Only* fails for all digit pairs, *IMMA* only succeeds in one pair, and *Opt $\kappa$* fails for 32 out of 90 pairs. In contrast, our method achieves success across all digit pairs demonstrating its effectiveness for immunization.

Thus far, we have conducted experiments strictly following the immunization setting that we have proposed in Sec. 3. However, one limitation of the setting is that the feature extractor is assumed to be linear, which limits its real-world potential. To further study the practicality of our method, despite the theoretical gap, we conduct experiments with non-linear models, *i.e.*, deep-nets, on a larger-scale image classification dataset of ImageNet.

**5.2. Experiments on Immunizing Deep-Nets**

**Immunization task.** In this experiment, we consider a common setup of linear probing on models pre-trained on ImageNet (Deng et al., 2009), *i.e.*, ImageNet serves as $\mathcal{D}_\text{P}$. For $\mathcal{D}_\text{H}$ we experiment with the Stanford Cars Dataset (Krause et al., 2013) and Country211 Dataset (Radford et al., 2021). These datasets have been previously used for studying transfer learning (Radford et al., 2021) for image classification. More dataset details are deferred to Appendix C.1.

**Experiment setup.** For non-linear models, we experiment with the architecture of ResNet18 (He et al., 2016) and ViT (Dosovitskiy, 2021). Here we study a practical setting where a given model with parameters $\theta_0$ has already been trained on $\mathcal{D}_\text{P}$ and would undergo immunization to obtain $\theta_\text{I}$ to be released to the public.

Note that as we are now using an initialization of $\theta_0$ and a non-linear feature extractor $f_\theta$, we extend the RIR metric to consider those changes. Specifically, we propose

$$\text{RIR}_{\theta_0} \triangleq \underbrace{\left( \frac{\kappa(\tilde{\boldsymbol{H}}_\text{H}(\theta_\text{I}))}{\kappa(\tilde{\boldsymbol{H}}_\text{H}(\theta_0))} \right)}_{\text{(i)}} \Big/ \underbrace{\left( \frac{\kappa(\tilde{\boldsymbol{H}}_\text{P}(\theta_\text{I}))}{\kappa(\tilde{\boldsymbol{H}}_\text{P}(\theta_0))} \right)}_{\text{(ii)}} \qquad (17)$$

where we compare the immunized model $\theta_\text{I}$ relative to the

initialization model $\theta_0$. Here, $\tilde{\boldsymbol{H}}(\theta)$ denotes the Hessian for linear probing on $\mathcal{D}_\text{H}$ with a non-linear $f_\theta$, *i.e.*,

$$\tilde{\boldsymbol{H}}_\text{H}(\theta) = \nabla_\mathbf{w}^2 \mathcal{L}(\mathcal{D}_\text{H}, \mathbf{w}, \theta) = \tilde{\boldsymbol{X}}_\text{H}(\theta)^\top \tilde{\boldsymbol{X}}_\text{H}(\theta). \qquad (18)$$

Here, $\tilde{\boldsymbol{X}}_\text{H}(\theta) \triangleq [f_\theta(\boldsymbol{x}); \forall \boldsymbol{x} \in \mathcal{D}_\text{H}] \in \mathbb{R}^{N \times D_\text{hid}}$ denotes the concatenation of the features, with dimensions $D_\text{hid}$, extracted from the input data. Due to memory constraints, we approximate Eq. (17) by randomly sampling 20 groups of training data, each containing 100 samples, and reporting the average values.

Finally, we also report the task performance after immunization. This is because, as the feature extractor is non-linear we are no longer guaranteed to retain the task performance. For ResNet18, we immunize only the last two convolutional blocks of the trained feature extractor and keep the rest of the parameters frozen as in $\theta_0$. For ViT, we only immunize the final transformer block. We optimize Eq. (11) using SGD with momentum, the default optimizer on ImageNet. Further details are provided in Appendix C.2.

**Results.** We present the quantitative results of immunizing deep-nets in Tab. 3. On both Cars and Country211 datasets, our method demonstrates strong performance when applied to ResNetg18 and ViT, as indicated by $\text{RIR}_{\theta_0} \gg 1$. In comparison, $\mathcal{R}_\text{ill}$ *Only* and *IMMA* did not effectively immunize the models in all evaluated settings. Next, *Opt $\kappa$* also succeeds in immunizing the models but our proposed method outperforms it in $\text{RIR}_{\theta_0}$.

Next, we report the test accuracy of the immunized models on $\mathcal{D}_\text{P}$, *i.e.*, ImageNet1K. On the ResNet18 architecture, we observe a reduction in test-accuracy from the initialization model $\theta_0$ of 68.24% to 62.36% when $\mathcal{D}_\text{H}$ is Cars and 65.01% when $\mathcal{D}_\text{H}$ is Country211. Interestingly, on the ViT architecture the test-accuracy *increased* from 81.78% to 82.79% for Cars, and 83.17% for Country211. These results suggested that it is possible to immunize a non-linear model against the harmful task without losing the effectiveness of the other task.

*Table 3.* Quantitative results of immunization of model pre-trained on ImageNet (Deng et al., 2009), computed over 3 random seeds. The $\mathcal{D}_\text{P}$ test accuracy for the off-the-shelf model initialization of $\theta_0$ on ResNet18 is 68.24% and that of ViT is 81.78%. We report $\texttt{RIR}_{\theta_0}$ to measure the quality of immunization. Test accuracy of $\mathcal{D}_\text{P}$ is reported to ensure the performance on the pre-training task is maintained.

| $\mathcal{D}_\text{H}$ | Method | ResNet18 | | | | ViT | | | |
|---|---|---|---|---|---|---|---|---|---|
| | | Eq. (17) (i)↑ | Eq. (17) (ii)↓ | $\texttt{RIR}_{\theta_0}$ ↑ | $\mathcal{D}_\text{P}$ Test Acc. (%) ↑ | Eq. (17) (i)↑ | Eq. (17) (ii)↓ | $\texttt{RIR}_{\theta_0}$ ↑ | $\mathcal{D}_\text{P}$ Test Acc. (%) ↑ |
| | Init. $\theta_0$ | 1.0 | 1.0 | 1.0 | 68.24 | 1.0 | 1.0 | 1.0 | 81.78 |
| Cars | $\mathcal{R}_{111}$ *Only* | 1.878 ±0.034 | 1.786 ±0.025 | 1.057 ±0.026 | **63.84** ±0.292 | **13.121** ±0.038 | 4.097 ±0.098 | 3.342 ±0.048 | 82.21 ±0.035 |
| | *IMMA* | 0.866 ±0.002 | 0.889 ±0.001 | 0.974 ±0.002 | 63.57 ±0.234 | 1.422 ±0.006 | 2.090 ±0.043 | 0.702 ±0.007 | 81.89 ±0.010 |
| | *Opt κ* | 1.217 ±0.021 | 0.798 ±0.005 | 1.527 ±0.019 | 63.65 ±0.148 | 3.598 ±0.510 | **0.171** ±0.033 | 26.369 ±2.814 | 82.51 ±0.085 |
| | **Ours** | 2.386 ±0.442 | **0.699** ±0.062 | **3.467** ±0.358 | 62.36 ±0.173 | 7.945 ±0.247 | 0.323 ±0.086 | **34.517** ±0.886 | **82.79** ±0.200 |
| Country211 | $\mathcal{R}_{111}$ *Only* | **20.727** ±0.791 | 20.675 ±1.685 | 1.038 ±0.05 | 62.17 ±1.599 | 69.291 ±1.198 | 63.519 ±6.62 | 1.122 ±0.097 | 80.73 ±0.129 |
| | *IMMA* | 0.791 ±0.005 | 0.814 ±0.006 | 0.972 ±0.007 | **67.03** ±0.146 | 6.242 ±0.203 | 7.599 ±0.717 | 0.845 ±0.048 | 82.47 ±0.036 |
| | *Opt κ* | 1.538 ±0.155 | 1.053 ±0.091 | 1.472 ±0.043 | 66.81 ±0.115 | 4.589 ±0.079 | **0.300** ±0.106 | 16.498 ±5.183 | 82.79 ±0.023 |
| | **Ours** | 3.287 ±0.33 | **0.399** ±0.034 | **8.714** ±0.672 | 65.01 ±0.143 | 20.894 ±1.425 | 0.700 ±0.082 | **41.341** ±0.967 | **83.17** ±0.075 |

To further show a larger Eq. (17) (i) indicating that a model is better immunized, we report the linear probed (fine-tuned) results on different feature extractors and provide the test accuracy on $\mathcal{D}_\text{H}$, where $\mathcal{D}_\text{H}$ is the Stanford Cars dataset. As shown in Fig. 3, our method exhibits the slowest convergence rate on both ResNet18 and ViT, indicated by the lowest test accuracy compared with baselines. In summary, our method remains effective on deep-nets, producing models that satisfy the requirements of an immunized model as in Definition 3.1.

# 6. Related Work

We briefly discuss related research on AI safety and the condition number.

**AI safety, model un/re-learning, and immunization.** AI safety has received attention lately, specifically in generative AI, due to the impressive progress. We refer the reader to Brundage et al. (2018); Marchal et al. (2024); Bengio et al. (2025) for a more in-depth discussion on this topic. In the following, we will discuss model unlearning, one of the ways to mitigate the potential of misuse, followed by model immunization, which protects a model against relearning.

Machine unlearning was first introduced by Cao & Yang (2015) to remove a user's private information from a model. Approximate unlearning aims to achieve this by modifying the pre-trained model directly using the specific data samples to erase, without requiring full retraining (Nguyen et al., 2020; Wu et al., 2022; Guo et al., 2019; Sekhari et al., 2021; Neel et al., 2021). In the context of text-to-image models, several methods for concept erasure have been proposed. These include inference-time approaches (Brack et al., 2023; Schramowski et al., 2023), fine-tuning of diffusion models (Gandikota et al., 2023; Kim et al., 2023; Kumari et al., 2023), and direct model editing (Zhang et al., 2024; Gandikota et al., 2024).

While promising, these works still face potential risks of the re-emergence/re-learning of harmful data (Zheng & Yeh, 2024; Zheng et al., 2024; Zhan et al., 2024; Bertran et al., 2024; Xu et al., 2025). To avoid relearning or further fine-tuning on harmful data, Zheng & Yeh (2024) propose to immunize the text-to-image models against malicious fine-tuning and Zheng & Yeh (2025) extend model immunization to multi-concept settings. Recent work highlights the importance of preventing re-finetuning or distillation on harmful tasks in language models (Huang et al., 2024; Savani et al., 2025; Rosati et al., 2024a; Tamirisa et al., 2024; Rosati et al., 2024b; Henderson et al., 2023) and encoder probing (Ding et al., 2025), which is closely related to our goal. While we also study the task of model immunization, different from Zheng & Yeh (2024) that primarily focuses on empirical applications on generative tasks, our work aims to provide a more principled understanding of model immunization by analyzing it through the lens of the condition number.

**Minimizing Condition Number.** Condition number has been a key factor in the convergence rates and accuracies of iterative methods, *e.g.*, Jacobi method (Arioli & Romani, 1985), steepest descent (Luenberger et al., 1984), conjugate gradient (Hestenes et al., 1952), for solving optimization problems from classic linear systems (Saad, 2003) to those with general nonlinear objectives (Nesterov, 2018) concerning modern machine learning applications. It is widely observed that a small condition number tends to speed up convergence and improve accuracy whereas a large condition number could lead to an unstable optimization procedure (Saarinen et al., 1993; Kress, 2012; Bengio et al., 2017; Guille-Escuret et al., 2021).

As a result, methods to minimize the condition number in various contexts have been proposed. Preconditioning (Evans, 1968), a technique that involves finding a matrix, *i.e.*, the preconditioner, to multiply with the original matrix, resulting in a new matrix with a significantly smaller condition number, is widely used for solving linear systems. The preconditioner can be constructed using methods

Fine-tuning accuracy with ResNet-18

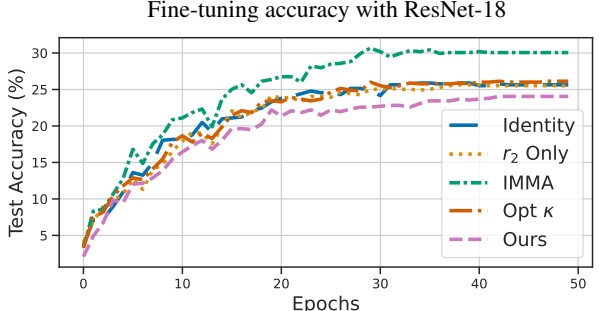

Fine-tuning accuracy with ViT

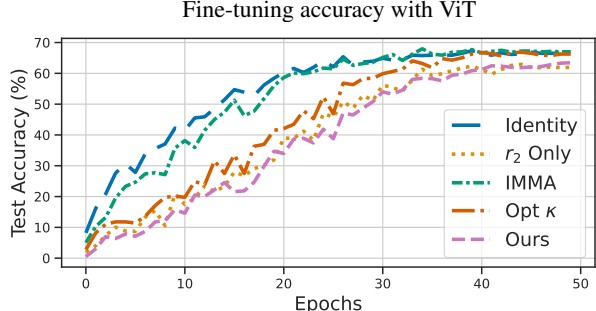

*Figure 3.* Test accuracy *vs.* Fine-tuning Epochs on $\mathcal{D}_\mathrm{H}$. We visualize the test accuracy of linear probing on ImageNet of different immunized models using gradient descent. Here $\mathcal{D}_\mathrm{H}$ is the Stanford Cars dataset.

such as semidefinite programming (Jambulapati et al., 2020; 2023; Qu et al., 2024) or matrix equilibration (Van der Sluis, 1969), and has recently found applications in deep learning (Saratchandran et al., 2024).

Most related to this work, Balazs et al. (2024) propose to regularize the condition number of weight matrices by directly adding the condition number term into the optimization objective and applying (sub)gradient descent. Observing that the condition number is discontinuous and nonconvex, Nenov et al. (2024) proposed a differentiable regularizer that minimizes the matrix condition number with a monotonic decrease guarantee if optimized with gradient descent. To the best of our knowledge, no notable effort has been made to increase or maximize the condition number.

## 7. Conclusion

We propose a framework for studying model immunization through the condition number of the Hessian matrix. We show that immunization can be achieved by increasing the condition number of harmful datasets while keeping it stable for the pre-training task. To achieve this, we introduce two differentiable regularizers and propose an algorithm that incorporates these regularizers into a gradient-based optimization algorithm. Empirical results on both linear and deep models demonstrate the effectiveness of our approach to model immunization. We believe that our proposed framework is a first step towards a more principled understanding of model immunization and will ultimately make open-sourced models safer.

## Acknowledgements

This project is supported in part by an NSF Award #2420724 and the Ross-Lynn Research Scholar Grant.

## Impact Statement

This paper presents work whose goal is to advance the field of Machine Learning and Optimization. While there are many potential societal consequences of our work, we believe that the benefits outweigh the harms. Specifically, the topic of model immunization is towards making AI safer.

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

# Appendix

The appendix is organized as follows:

- In Sec. A, we provide the complete statements of the properties of $\mathcal{R}_{\mathtt{well}}(S)$ for minimizing the condition number.
- In Sec. B, we provide the complete proof for the Theorems stated in the main paper.
- In Sec. C, we provide additional experiment details. The code will be open-sourced upon the acceptance of this paper.

## A. Properties of the Condition Number Minimizing Regularizer

**Theorem A.1** (Properties of $\kappa$-minimizing regularizer $\mathcal{R}_{\mathtt{well}}(S)$, Theorem 2.1, 2.2, 3.1, 3.2 in Nenov et al. (2024))**.**

*(1) [Nonnegativity]* $\forall\ S \in \mathbb{R}^{p_r \times p_c}$, $\mathcal{R}_{\mathtt{well}}(S) \geq 0$. *If* $S \neq \mathbf{0}$, $\mathcal{R}_{\mathtt{well}}(S) = 0$ *if and only if* $S$ *has full rank and* $\kappa(S) = 1$.

*(2) [Upper Bound]* $\kappa(S) \leq e^{p(\sigma_S^{\mathtt{min}})^{-2}\mathcal{R}_{\mathtt{well}}(S)}$, *i.e.,* $r(S)$ *is an upper bound of* $\log(\kappa(S))$ *as long as* $\sigma_S^{\mathtt{min}}$ *is bounded away from 0.*

*(3) [Differentiability]* *If* $\sigma_S^{\mathtt{max}} = \sigma_1 > \sigma_i$ *for any* $i > 1$, *i.e.,* $\sigma_S^{\mathtt{max}}$ *is unique, then* $\mathcal{R}_{\mathtt{well}}(S)$ *is differentiable and its gradient is given by* $\nabla_S \mathcal{R}_{\mathtt{well}}(S) = \sigma_1 u_1 v_1^\top - \frac{1}{p}S$.

*(4) [Monotonic Decrease]* *If* $\sigma_S^{\mathtt{max}}$ *is unique, update* $S$ *with* $\nabla_S \mathcal{R}_{\mathtt{well}}(S)$ *such that* $S' = S - \eta_1 \nabla_S \mathcal{R}_{\mathtt{well}}(S)$ *for* $0 < \eta_1 < \frac{\kappa(S)-1}{\left(1-\frac{1}{p}\right)\kappa(S)+\frac{1}{p}}$, *then* $\kappa(S') < \kappa(S)$.

## B. Proof of Propositions and Theorems

### B.1. Proof of Proposition 3.2.

**Proposition 3.2.** *The singular values of the Hessian matrix in Eq. (9) are given by*

$$\sigma_i = \sum_{j=1}^{D_{\mathtt{in}}} \left(\sigma_{\theta,i}(u_{\theta,i}^\top q_j)\sqrt{\gamma_j}\right)^2, \quad \forall i \in \{1, \ldots, D^{\mathtt{in}}\}. \tag{10}$$

*Here,* $\sigma_{\theta,i}$ *and* $u_{\theta,i}$ *correspond to the $i$-th singular value and vector of* $\theta$. *Next,* $\gamma_j$ *and* $q_j$ *correspond to the $j$-th singular value and vector of the covariance* $K$.

*Proof.* Substitute the SVD of $\theta$ and the eigendecomposition of $K$ into $\theta^\top K \theta$:

$$\theta^\top K \theta = (U_\theta \Sigma_\theta V_\theta^\top)^\top (Q \Gamma^2 Q^\top)(U_\theta \Sigma_\theta V_\theta^\top).$$

Simplify the expression:

$$\theta^\top K \theta = V_\theta (\Sigma_\theta U_\theta^\top Q \Gamma^2 Q^\top U_\theta \Sigma_\theta) V_\theta^\top.$$

Define $M = \Sigma_\theta U_\theta^\top Q \Gamma$, so that:

$$\theta^\top K \theta = V_\theta (M M^\top) V_\theta^\top.$$

The elements of $M$ are:

$$M[i,j] = \sigma_{\theta,i}(u_{\theta,i}^\top q_j)\gamma_j,$$

where $\sigma_{\theta,i}$'s for $i \in [d]$ are the singular values of $\theta$, $\gamma_j$'s for $i \in [d]$ are the diagonal entries of $\Gamma$, and $(u_{\theta,i}^\top q_j)$ measures the alignment between the $i$-th column of $U_\theta$ and the $j$-th column of $Q$.

We observe the following decomposition of $M$ in to two matrices $O$ and $D$:

$$
M = \begin{bmatrix}
\ddots & \vdots & \cdot^{\cdot^{\cdot}} \\
\cdots & \sigma_{\theta,i}(u_{\theta,i}^\top q_j)\gamma_j & \cdots \\
\cdot^{\cdot^{\cdot}} & \vdots & \ddots
\end{bmatrix}
$$

$$
= \begin{bmatrix}
\ddots & \vdots & \cdot^{\cdot^{\cdot}} \\
\cdots & \dfrac{\sigma_{\theta,i}(u_{\theta,i}^\top q_j)\gamma_j}{\sqrt{\sum_{j'}(\sigma_{\theta,i}(u_{\theta,i}^\top q_{j'})\gamma_{j'})^2}} & \cdots \\
\cdot^{\cdot^{\cdot}} & \vdots & \ddots
\end{bmatrix}
\begin{bmatrix}
\ddots & 0 & 0 \\
0 & \sqrt{\sum_{j'}(\sigma_{\theta,i}(u_{\theta,i}^\top q_{j'}\gamma_j)^2} & 0 \\
0 & 0 & \ddots
\end{bmatrix}
$$

$$
= OD
$$

where $O$ is an orthonormal matrix, i.e., $O^\top O = I$, and $D = \mathrm{diag}(d_1, \ldots, d_d)$ with $d_i = \sqrt{\sum_{j'}(\sigma_{\theta,i}(u_{\theta,i}^\top q_{j'})\gamma_{j'})^2}$ is a diagonal matrix. As a result, diagonal entries of $D^2$ are:

$$
d_i^2 = \sum_{j=1}^d \left(\sigma_{\theta,i}(u_{\theta,i}^\top q_j)\gamma_j\right)^2.
$$

Thus, $MM^\top = (OD)(OD)^\top = OD^2O^\top$, and the eigenvalues of $\theta^\top K\theta$ are the diagonal entries of $D^2$, given by:

$$
\sigma_i = d_i^2 = \sum_{j=1}^d \left(\sigma_{\theta,i}(u_{\theta,i}^\top q_j)\gamma_j\right)^2, \quad i = 1, \ldots, d.
$$

$\square$

## B.2. Proof of Theorem 4.1

B.2.1. PROOF OF THEOREM 4.1 (1)

**Theorem 4.1.** *(1) For any $S \in \mathbb{R}^{p_r \times p_c}$, $\mathcal{R}_{\mathtt{ill}}(S) \geq 0$, and $\mathcal{R}_{\mathtt{ill}}(S) = 0$ if and only if $\kappa(S) = \infty$.*

*Proof.* By definition, $\mathcal{R}_{\mathtt{ill}}(S) = \frac{1}{\frac{1}{2k}\|S\|_F^2 - \frac{1}{2}(\sigma_S^{\mathtt{min}})^2}$. Denote $\mathcal{R}'_{\mathtt{ill}}(S) = \frac{1}{2}\left((\sigma_S^{\mathtt{min}})^2 - \frac{1}{k}\|S\|_F^2\right)$, then we have $\mathcal{R}_{\mathtt{ill}}(S) = \frac{1}{-\mathcal{R}'_{\mathtt{ill}}(S)}$, and

$$
\mathcal{R}'_{\mathtt{ill}}(S) = \frac{1}{2}\left(\sigma_k^2 - \frac{1}{k}\sum_{i=1}^k \sigma_i^2\right)
$$

$$
= \frac{1}{2k}\sum_{i=1}^k \left(\sigma_k^2 - \sigma_i^2\right)
$$

$$
\leq 0,
$$

since $\forall\, i \in [k]$, $\sigma_S^{\mathtt{min}} = \sigma_k \leq \sigma_i$. As a result, $-\mathcal{R}'_{\mathtt{ill}}(S) \geq 0$ and $\mathcal{R}_{\mathtt{ill}}(S) = \frac{1}{-\mathcal{R}'_{\mathtt{ill}}(S)} \geq 0$, i.e., $\mathcal{R}_{\mathtt{ill}}(S)$ is non-negative.

Also, by definition, $\sigma_1 = \kappa(\boldsymbol{S})\sigma_k$. Therefore,

$$
\begin{aligned}
\mathcal{R}_{\text{ill}}(\boldsymbol{S}) &= \frac{2}{\frac{1}{k}\sum_{i=1}^{k}\sigma_i^2 - (\sigma_{\boldsymbol{S}}^{\min})^2} \\
&\leq \frac{2}{\frac{1}{k}\sigma_1^2 + \frac{k-1}{k}(\sigma_{\boldsymbol{S}}^{\min})^2 - (\sigma_{\boldsymbol{S}}^{\min})^2} \\
&= \frac{2}{\frac{1}{k}\left(\sigma_1^2 - (\sigma_{\boldsymbol{S}}^{\min})^2\right)} \\
&= \frac{2k}{(\kappa(\boldsymbol{S})^2 - 1)(\sigma_{\boldsymbol{S}}^{\min})^2}.
\end{aligned}
$$

If $\kappa(\boldsymbol{S}) = \infty$, $\mathcal{R}_{\text{ill}}(\boldsymbol{S}) \leq \frac{2k}{(\kappa(\boldsymbol{S})^2-1)(\sigma_{\boldsymbol{S}}^{\min})^2} = 0$ for $\sigma_{\boldsymbol{S}}^{\min} > 0$, which yields $\mathcal{R}_{\text{ill}}(\boldsymbol{S}) = 0$ given that $\mathcal{R}_{\text{ill}}(\boldsymbol{S}) \geq 0$.

Similarly, we have

$$
\begin{aligned}
\mathcal{R}_{\text{ill}}(\boldsymbol{S}) &= \frac{2}{\frac{1}{k}\sum_{i=1}^{k}\sigma_i^2 - (\sigma_{\boldsymbol{S}}^{\min})^2} \\
&\geq \frac{2}{\frac{k-1}{k}\sigma_1^2 + \frac{1}{k}(\sigma_{\boldsymbol{S}}^{\min})^2 - (\sigma_{\boldsymbol{S}}^{\min})^2} \\
&= \frac{2}{\frac{k-1}{k}\left(\sigma_1^2 - (\sigma_{\boldsymbol{S}}^{\min})^2\right)} \\
&= \frac{\frac{2k}{k-1}}{(\kappa(\boldsymbol{S})^2 - 1)(\sigma_{\boldsymbol{S}}^{\min})^2}.
\end{aligned}
$$

If $\mathcal{R}_{\text{ill}}(\boldsymbol{S}) = 0$, we have $\kappa(\boldsymbol{S}) \geq \sqrt{\frac{\frac{2k}{k-1}}{\mathcal{R}_{\text{ill}}(\boldsymbol{S})(\sigma_{\boldsymbol{S}}^{\min})^2} + 1} = \infty$ which yields $\kappa(\boldsymbol{S}) = \infty$. $\qquad\square$

### B.2.2. PROOF OF THEOREM 4.1 (2)

To prove Theorem 4.1 (2), we start by analyzing $\mathcal{R}'_{\text{ill}}(\boldsymbol{S}) = \frac{1}{2}\left((\sigma_{\boldsymbol{S}}^{\min})^2 - \frac{1}{k}\|\boldsymbol{S}\|_F^2\right)$ with the following lemma.

**Lemma B.1.** *For* $\mathcal{R}'_{\text{ill}}(\boldsymbol{S}) = \frac{1}{2}\left((\sigma_{\boldsymbol{S}}^{\min})^2 - \frac{1}{k}\|\boldsymbol{S}\|_F^2\right)$,

$$
\frac{1}{\kappa(\boldsymbol{S})} \leq e^{\frac{k}{k-1}\sigma_1^{-2}\mathcal{R}'_{\text{ill}}(\boldsymbol{S})} \tag{19}
$$

*That is,* $\mathcal{R}'_{\text{ill}}(\boldsymbol{S})$ *is an upper bound of* $\log\left(\frac{1}{\kappa(\boldsymbol{S})}\right)$, *i.e.,* $-\log(\kappa(\boldsymbol{S}))$.

*Proof.* Similar to the proof of Theorem 3.2 in (Nenov et al., 2024),

$$
\begin{aligned}
2\mathcal{R}'_{\text{ill}}(\boldsymbol{S}) &= (\sigma_{\boldsymbol{S}}^{\min})^2 - \frac{1}{k}\|\boldsymbol{S}\|_F^2 \\
&= (\sigma_{\boldsymbol{S}}^{\min})^2 - \frac{1}{k}\sum_{i=1}^{k}\sigma_i^2 \\
&\geq (\sigma_{\boldsymbol{S}}^{\min})^2 - \frac{1}{k}\left((k-1)\sigma_1^2 + (\sigma_{\boldsymbol{S}}^{\min})^2\right) \\
&= \left(1 - \frac{1}{k}\right)\left((\sigma_{\boldsymbol{S}}^{\min})^2 - \sigma_1^2\right)
\end{aligned}
$$

In the meantime,

$$2\log\left(\frac{1}{\kappa(\boldsymbol{S})}\right) = -\log\left(\kappa(\boldsymbol{S})^2\right)$$
$$= \log\left(\left(\sigma_{\boldsymbol{S}}^{\mathtt{min}}\right)^2\right) - \log\left(\sigma_1^2\right)$$
$$\leq -\frac{1}{\sigma_1^2}\left(\sigma_1^2 - \left(\sigma_{\boldsymbol{S}}^{\mathtt{min}}\right)^2\right)$$
$$= \frac{1}{\sigma_1^2}\left(\left(\sigma_{\boldsymbol{S}}^{\mathtt{min}}\right)^2 - \sigma_1^2\right)$$

in which the inequality follows from the Mean Value Theorem. As a result,

$$\frac{1}{\kappa(\boldsymbol{S})} \leq e^{\frac{1}{2\sigma_1^2}\left(\left(\sigma_{\boldsymbol{S}}^{\mathtt{min}}\right)^2 - \sigma_1^2\right)}$$
$$\leq e^{\frac{1}{2\sigma_1^2}\left(\frac{k}{k-1}2\mathcal{R}_{\mathtt{ill}}'(\boldsymbol{S})\right)}$$
$$= e^{\frac{k}{k-1}\sigma_1^{-2}\mathcal{R}_{\mathtt{ill}}'(\boldsymbol{S})}$$

$\square$

**Theorem 4.1.** *(2)* $\frac{1}{\log(\kappa(\boldsymbol{S}))} \leq (\sigma_{\boldsymbol{S}}^{\mathtt{max}})^2 \mathcal{R}_{\mathtt{ill}}(\boldsymbol{S})$, *i.e.,* $\mathcal{R}_{\mathtt{ill}}(\boldsymbol{S})$ *upper bounds* $\frac{1}{\log(\kappa(\boldsymbol{S}))}$ *when* $\sigma_{\boldsymbol{S}}^{\mathtt{max}}$ *is reasonably away from* $\infty$.

*Proof.* Taking the logarithm of Lemma B.1, we have

$$-\log\left(\kappa\left(\boldsymbol{S}\right)\right) \leq \frac{k}{k-1}\sigma_1^{-2}\mathcal{R}_{\mathtt{ill}}'\left(\boldsymbol{S}\right).$$

Negating both sides,

$$\log\left(\kappa\left(\boldsymbol{S}\right)\right) \geq -\frac{k}{k-1}\sigma_1^{-2}\mathcal{R}_{\mathtt{ill}}'\left(\boldsymbol{S}\right).$$

Finally, taking the reciprocal,

$$\frac{1}{\log\left(\kappa\left(\boldsymbol{S}\right)\right)} \leq \frac{k-1}{k}\frac{\sigma_1^2}{-\mathcal{R}_{\mathtt{ill}}'\left(\kappa\left(\boldsymbol{S}\right)\right)}$$
$$= \frac{k-1}{k}\frac{\sigma_1^2}{\frac{1}{2}\left(\sigma_{\boldsymbol{S}}^{\mathtt{min}}\right)^2 - \frac{1}{2k}\|\boldsymbol{S}\|_F^2}$$
$$\leq \sigma_1^2\mathcal{R}_{\mathtt{ill}}\left(\boldsymbol{S}\right)$$

$\square$

### B.2.3. PROOF OF THEOREM 4.1 (3)

To analyze the differentiability of $\mathcal{R}_{\mathtt{ill}}(\boldsymbol{S}) = \frac{1}{\frac{1}{2k}\|\boldsymbol{S}\|_F^2 - \frac{1}{2}\left(\sigma_{\boldsymbol{S}}^{\mathtt{min}}\right)^2}$, we start by analyzing the differentiability of $\mathcal{R}_{\mathtt{ill}}'(\boldsymbol{S}) = \frac{1}{2}\left(\left(\sigma_{\boldsymbol{S}}^{\mathtt{min}}\right)^2 - \frac{1}{k}\|\boldsymbol{S}\|_F^2\right)$, which needs the following lemma as a prerequisite.

**Lemma B.2** (Theorem 3.1 in (Lewis, 1995) without Convexity). *If a function* $f : \mathbb{R}^p \to \mathbb{R}$ *is absolutely symmetric, that is,* $\forall\, \boldsymbol{x} \in \mathbb{R}^p$ *and any* $\boldsymbol{y}$ *as a permutation of* $\boldsymbol{x}$, $f(\boldsymbol{x}) = f(\boldsymbol{y})$, *then* $f \circ \boldsymbol{\sigma}$ *is differentiable at matrix* $\boldsymbol{S} \in \mathbb{R}^{p_1 \times p_2}$ *if and only if* $f$ *is differentiable at* $\boldsymbol{\sigma} = \boldsymbol{\sigma}(\boldsymbol{S})$. *In this case, for the singular value decomposition* $\boldsymbol{S} = \boldsymbol{U}\mathrm{Diag}(\boldsymbol{\sigma})\boldsymbol{V}^\top$,

$$\nabla\left(f \circ \boldsymbol{\sigma}\right)\left(\boldsymbol{S}\right) = \boldsymbol{U}\mathrm{Diag}(\nabla f(\boldsymbol{\sigma}))\boldsymbol{V}^\top.$$

*Proof.* For the forward direction, by Corollary 2.5 in (Lewis, 1995), for $S = U\text{Diag}(\sigma)V^\top$,

$$\partial (f \circ \sigma)(S) = \left\{ U\text{Diag}(\mu)V^\top \Big| \mu \in \partial f(\sigma) \right\}.$$

By Theorem 25.1 in (Rockafellar, 1970), since $f \circ \sigma$ is differentiable at matrix $S \in \mathbb{R}^{p_1 \times p_2}$, we know that its subgradient $\partial (f \circ \sigma)(S)$ is a singleton, meaning that $U\text{Diag}(\mu)V^\top$ is unique, and consequently, $\mu \in \partial f(\sigma)$ is unique. As a result, $\partial f(\sigma)$ is also a singleton, which, again by Corollary 2.5 in (Lewis, 1995), indicates that $f$ is differentiable at $\sigma$. The reverse direction holds true following a similar argument. $\qquad\square$

**Lemma B.3.** *For $S = U\text{Diag}(\sigma)V^\top$, in which $\sigma = [\sigma_1, \cdots, \sigma_k]^\top$ such that $\sigma_S^{\mathtt{max}} = \sigma_1 \geq \sigma_2 \geq \cdots > \sigma_k = \sigma_S^{\mathtt{min}}$, i.e., $\sigma_k < \sigma_i$ for any $i < k$, $\mathcal{R}'_{\mathtt{ill}}(S) = \frac{1}{2}\left((\sigma_S^{\mathtt{min}})^2 - \frac{1}{k}\|S\|_F^2\right)$ is differentiable and for $u_k$, $v_k$ as the $k^{th}$ column vector of $U$, $V$,*

$$\nabla \mathcal{R}'_{\mathtt{ill}}(S) = \sigma_S^{\mathtt{min}} u_k v_k^\top - \frac{1}{k}S. \tag{20}$$

*Proof.* For $x \in \mathbb{R}^k$, denote

$$\mathcal{R}'_{\mathtt{ill},1}(x) = \min_{i \in [k]} \frac{1}{2}x_i^2, \quad \mathcal{R}'_{\mathtt{ill},2}(x) = \frac{1}{2k}\sum_{i=1}^{k} x_i^2.$$

With $\mathcal{R}'_{\mathtt{ill}}(S) = \frac{1}{2}(\sigma_S^{\mathtt{min}})^2 - \frac{1}{2k}\|S\|_F^2$, we first analyze $\frac{1}{2}(\sigma_S^{\mathtt{min}})^2$. By the subdifferential of piecewise minimum given by Proposition 4.9 in (Mordukhovich, 2018), we have for $x \in \mathbb{R}^k$,

$$\partial_x \mathcal{R}'_{\mathtt{ill},1}(x) \subset \left\{ \partial_x \left(\frac{1}{2}x_i^2\right) \Big| i \in \arg\min_{j \in [k]} \frac{1}{2}x_j^2 \right\}$$

$$= \left\{ x_i e_i \Big| i \in \arg\min_{j \in [k]} \frac{1}{2}x_j^2 \right\}$$

$$= \left\{ x_i e_i \Big| i \in \arg\min_{j \in [k]} |x_j| \right\}$$

in which $e_i$ is the $i^{th}$ vector from the $k$-dimensional standard basis. Therefore,

$$\partial_\sigma \mathcal{R}'_{\mathtt{ill},1}(\sigma) \subset \left\{ \sigma_i e_i \Big| i \in \arg\min_{j \in [k]} \sigma_j \right\}$$

Since for any $i < k$, $\sigma_k < \sigma_i$, i.e., the minimum non-zero singular value $\sigma_S^{\mathtt{min}}$ is unique, we know that the subdifferential $\left\{ \sigma_i e_i \Big| i \in \arg\min_{j \in [k]} \sigma_j \right\} = \{\sigma_S^{\mathtt{min}}\}$ is a singleton. Therefore, by Theorem 25.1 in (Rockafellar, 1970), we know $\mathcal{R}'_{\mathtt{ill},1}$ is differentiable with respect to $\sigma$ and $\nabla_\sigma \mathcal{R}'_{\mathtt{ill},1}(\sigma) = \sigma_S^{\mathtt{min}} e_k$. Regarding $\sigma = \sigma(S)$ as a function of $S$ in which $\sigma(\cdot)$ represents taking the singular values of a matrix, we have by Corollary 2.5 in (Lewis, 1995)

$$\partial_S \left(\frac{1}{2}(\sigma_S^{\mathtt{min}})^2\right) = \partial_S (\mathcal{R}'_{\mathtt{ill},1} \circ \sigma)(S)$$

$$= \left\{ U\text{Diag}(\mu)V^\top \Big| \mu \in \partial_\sigma \mathcal{R}'_{\mathtt{ill},1}(\sigma) \right\}$$

Given that $\mathcal{R}'_{\mathtt{ill},1}$ is differentiable and apparently also absolutely symmetric with respect to $\sigma$, by Lemma B.2, we know $\frac{1}{2}(\sigma_S^{\mathtt{min}})^2$ is also differentiable and

$$\nabla \left(\frac{1}{2}(\sigma_S^{\mathtt{min}})^2\right) = U\text{Diag}(\nabla_\sigma \mathcal{R}'_{\mathtt{ill},1}(\sigma))V^\top$$

$$= U\text{Diag}(\sigma_S^{\mathtt{min}} e_p)V^\top$$

$$= \sigma_S^{\mathtt{min}} u_k v_k^\top.$$

In addition, we have

$$\partial_{\boldsymbol{S}}\left(\frac{1}{2k}\|\boldsymbol{S}\|_F^2\right) = \partial_{\boldsymbol{S}}\left(\frac{1}{2k}\sum_{i=1}^{k}\boldsymbol{\sigma}\left(\boldsymbol{S}\right)^2\right)$$

$$= \left\{\boldsymbol{U}\mathrm{Diag}(\boldsymbol{\mu})\boldsymbol{V}^\top \,\Big|\, \boldsymbol{\mu} \in \partial_{\boldsymbol{\sigma}}\mathcal{R}'_{\mathrm{ill},2}(\boldsymbol{\sigma})\right\}$$

by Corollary 2.5 in (Lewis, 1995). $\mathcal{R}'_{\mathrm{ill},2}$ is apparently differentiable with $\nabla\mathcal{R}'_{\mathrm{ill},2}(\boldsymbol{x}) = \frac{1}{k}\boldsymbol{x}$. Therefore, again by Lemma B.2,

$$\nabla\left(\frac{1}{2k}\|\boldsymbol{S}\|_F^2\right) = \boldsymbol{U}\mathrm{Diag}(\nabla\mathcal{R}'_{\mathrm{ill},2}(\boldsymbol{\sigma}_S))\boldsymbol{V}^\top$$

$$= \frac{1}{k}\boldsymbol{U}\mathrm{Diag}\left(\boldsymbol{\sigma}_S\right)\boldsymbol{V}^\top$$

$$= \frac{1}{k}\boldsymbol{S}.$$

By the linearity of gradients,

$$\nabla\mathcal{R}'_{\mathrm{ill}}(\boldsymbol{S}) = \nabla\left(\frac{1}{2}\left(\sigma_{\boldsymbol{S}}^{\mathtt{min}}\right)^2\right) - \nabla\left(\frac{1}{2k}\|\boldsymbol{S}\|_F^2\right)$$

$$= \sigma_{\boldsymbol{S}}^{\mathtt{min}}\boldsymbol{u}_k\boldsymbol{v}_k^\top - \frac{1}{k}\boldsymbol{S},$$

which completes the proof. $\square$

**Theorem 4.1**. *(3) If $\sigma_{\boldsymbol{S}}^{\mathtt{min}} = \sigma_k < \sigma_i$ for any $i < k$, then $\mathcal{R}_{\mathrm{ill}}(\boldsymbol{S})$ is differentiable and $\nabla_{\boldsymbol{S}}\mathcal{R}_{\mathrm{ill}}(\boldsymbol{S}) = \frac{\sigma_k\boldsymbol{u}_k\boldsymbol{v}_k^\top - \frac{1}{k}\boldsymbol{S}}{\left(\frac{1}{2k}\|\boldsymbol{S}\|_F^2 - \frac{1}{2}\left(\sigma_{\boldsymbol{S}}^{\mathtt{min}}\right)^2\right)^2}.$*

*Proof.* Since $\mathcal{R}_{\mathrm{ill}}(\boldsymbol{S}) = \frac{1}{\frac{1}{2k}\|\boldsymbol{S}\|_F^2 - \frac{1}{2}\left(\sigma_{\boldsymbol{S}}^{\mathtt{min}}\right)^2}$, we have

$$\partial\mathcal{R}_{\mathrm{ill}}\left(\boldsymbol{S}\right) = \frac{-\partial\left(\frac{1}{2k}\|\boldsymbol{S}\|_F^2 - \frac{1}{2}\left(\sigma_{\boldsymbol{S}}^{\mathtt{min}}\right)^2\right)}{\left(\frac{1}{2k}\|\boldsymbol{S}\|_F^2 - \frac{1}{2}\left(\sigma_{\boldsymbol{S}}^{\mathtt{min}}\right)^2\right)^2}$$

$$= \frac{\partial\left(\frac{1}{2}\left(\sigma_{\boldsymbol{S}}^{\mathtt{min}}\right)^2 - \frac{1}{2k}\|\boldsymbol{S}\|_F^2\right)}{\left(\frac{1}{2k}\|\boldsymbol{S}\|_F^2 - \frac{1}{2}\left(\sigma_{\boldsymbol{S}}^{\mathtt{min}}\right)^2\right)^2}$$

$$= \frac{\partial\mathcal{R}'_{\mathrm{ill}}\left(\boldsymbol{S}\right)}{\left(\frac{1}{2k}\|\boldsymbol{S}\|_F^2 - \frac{1}{2}\left(\sigma_{\boldsymbol{S}}^{\mathtt{min}}\right)^2\right)^2}$$

By Lemma B.3, we know that if $\sigma_{\boldsymbol{S}}^{\mathtt{min}} = \sigma_k < \sigma_i$ for any $i < k$, $\mathcal{R}'_{\mathrm{ill}}\left(\boldsymbol{S}\right)$ is differentiable and $\nabla\mathcal{R}'_{\mathrm{ill}}(\boldsymbol{S}) = \sigma_{\boldsymbol{S}}^{\mathtt{min}}\boldsymbol{u}_k\boldsymbol{v}_k^\top - \frac{1}{k}\boldsymbol{S}$. Consequently, $\mathcal{R}_{\mathrm{ill}}(\boldsymbol{S})$ is differentiable and

$$\nabla\mathcal{R}_{\mathrm{ill}}\left(\boldsymbol{S}\right) = \frac{\sigma_{\boldsymbol{S}}^{\mathtt{min}}\boldsymbol{u}_k\boldsymbol{v}_k^\top - \frac{1}{k}\boldsymbol{S}}{\left(\frac{1}{2k}\|\boldsymbol{S}\|_F^2 - \frac{1}{2}\left(\sigma_{\boldsymbol{S}}^{\mathtt{min}}\right)^2\right)^2}.$$

$\square$

### B.2.4. PROOF OF THEOREM 4.1 (4)

**Theorem 4.1**. *(4) If $\sigma_{\boldsymbol{S}}^{\mathtt{min}}$ is unique, update $S$ with $\nabla_{\boldsymbol{S}}\mathcal{R}_{\mathrm{ill}}(\boldsymbol{S})$ such that $\boldsymbol{S}' = \boldsymbol{S} - \eta_2\nabla_{\boldsymbol{S}}\mathcal{R}_{\mathrm{ill}}(\boldsymbol{S})$ for $0 < \eta_2 < \frac{k}{k-1}\left(\frac{1}{2k}\|\boldsymbol{S}\|_F^2 - \frac{1}{2}\left(\sigma_{\boldsymbol{S}}^{\mathtt{min}}\right)^2\right)^2$, then $\kappa\left(\boldsymbol{S}'\right) > \kappa\left(\boldsymbol{S}\right)$.*

*Proof.* Given that $S' = S - \eta_2 \nabla \mathcal{R}_{\mathtt{ill}}(S)$ and that $\nabla \mathcal{R}_{\mathtt{ill}}(S) = \frac{\sigma_S^{\min} u_k v_k^\top - \frac{1}{k} S}{\left( \frac{1}{2k} \|S\|_F^2 - \frac{1}{2} (\sigma_S^{\min})^2 \right)^2} = \frac{1}{\mathcal{R}'_{\mathtt{ill}}(S)^2} \left( \sigma_k u_k v_k^\top - \frac{1}{k} S \right)$ for

$\mathcal{R}'_{\mathtt{ill}}(S) = \frac{1}{2} \left( \sigma_k^2 - \frac{1}{k} \|S\|_F^2 \right)$,

$$
\begin{aligned}
S' &= S - \eta_2 \nabla \mathcal{R}_{\mathtt{ill}}(S) \\
&= S - \frac{\eta_2}{\mathcal{R}'_{\mathtt{ill}}(S)^2} \left( \sigma_k u_k v_k^\top - \frac{1}{k} S \right) \\
&= \left( 1 + \frac{\eta_2}{k \mathcal{R}'_{\mathtt{ill}}(S)^2} \right) S - \frac{\eta_2}{\mathcal{R}'_{\mathtt{ill}}(S)^2} \sigma_k u_k v_k^\top \\
&= \left( 1 + \frac{\eta_2}{k \mathcal{R}'_{\mathtt{ill}}(S)^2} \right) \sum_{i=1}^k \sigma_i u_i v_i^\top - \frac{\eta_2}{\mathcal{R}'_{\mathtt{ill}}(S)^2} \sigma_k u_k v_k^\top \\
&= \left( 1 + \frac{\eta_2}{k \mathcal{R}'_{\mathtt{ill}}(S)^2} \right) \sum_{i=1}^{k-1} \sigma_i u_i v_i^\top + \left( 1 + \frac{\eta_2}{k \mathcal{R}'_{\mathtt{ill}}(S)^2} - \frac{\eta_2}{\mathcal{R}'_{\mathtt{ill}}(S)^2} \right) \sigma_k u_k v_k^\top \\
&= U \mathrm{Diag}(\sigma_{S'}) V^\top.
\end{aligned}
$$

where $\sigma_{S'} = \left[ \left( 1 + \frac{\eta_2}{k \mathcal{R}'_{\mathtt{ill}}(S)^2} \right) \sigma_1, \cdots, \left( 1 + \frac{\eta_2}{k \mathcal{R}'_{\mathtt{ill}}(S)^2} \right) \sigma_{k-1}, \left( 1 + \frac{\eta_2}{k \mathcal{R}'_{\mathtt{ill}}(S)^2} - \frac{\eta_2}{\mathcal{R}'_{\mathtt{ill}}(S)^2} \right) \sigma_k \right]^\top$ is the vector formed by the singular values of $S'$ but not necessarily in the decreasing order.

Now we argue that $\left( 1 + \frac{\eta_2}{k \mathcal{R}'_{\mathtt{ill}}(S)^2} \right) \sigma_1$ remains to be the maximum singular value while $\left( 1 + \frac{\eta_2}{k \mathcal{R}'_{\mathtt{ill}}(S)^2} - \frac{\eta_2}{\mathcal{R}'_{\mathtt{ill}}(S)^2} \right) \sigma_k$ the minimum. Since $\sigma_k < \sigma_i$ for any $i < k$, i.e., $\sigma_S^{\min} = \sigma_k$ is unique, we must have $0 < \beta < 1$ such that $\sigma_k = \beta \sigma_{k-1}$. Also, given that $\eta_2 < \frac{k}{k-1} \left( \frac{1}{2k} \|S\|_F^2 - \frac{1}{2} \sigma_k^2 \right)^2 = \frac{k \mathcal{R}'_{\mathtt{ill}}(S)^2}{k-1}$, we have $1 + \frac{\eta_2}{k \mathcal{R}'_{\mathtt{ill}}(S)^2} - \frac{\eta_2}{\mathcal{R}'_{\mathtt{ill}}(S)^2} > 0$. Therefore,

$$
\begin{aligned}
\left( 1 + \frac{\eta_2}{k \mathcal{R}'_{\mathtt{ill}}(S)^2} - \frac{\eta_2}{\mathcal{R}'_{\mathtt{ill}}(S)^2} \right) \sigma_k \\
= \left( 1 + \frac{\eta_2}{k \mathcal{R}'_{\mathtt{ill}}(S)^2} \right) \frac{1 + \frac{\eta_2}{k \mathcal{R}'_{\mathtt{ill}}(S)^2} - \frac{\eta_2}{\mathcal{R}'_{\mathtt{ill}}(S)^2}}{1 + \frac{\eta_2}{k \mathcal{R}'_{\mathtt{ill}}(S)^2}} \sigma_k \\
= \left( 1 + \frac{\eta_2}{k \mathcal{R}'_{\mathtt{ill}}(S)^2} \right) \left( 1 - \frac{\frac{\eta_2}{\mathcal{R}'_{\mathtt{ill}}(S)^2}}{1 + \frac{\eta_2}{k \mathcal{R}'_{\mathtt{ill}}(S)^2}} \right) \sigma_k \\
< \left( 1 + \frac{\eta_2}{k \mathcal{R}'_{\mathtt{ill}}(S)^2} \right) \sigma_k \\
< \left( 1 + \frac{\eta_2}{k \mathcal{R}'_{\mathtt{ill}}(S)^2} \right) \frac{1}{\beta} \sigma_k \\
= \left( 1 + \frac{\eta_2}{k \mathcal{R}'_{\mathtt{ill}}(S)^2} \right) \sigma_{k-1}.
\end{aligned}
$$

Since $\sigma_1 \geq \sigma_2(S) \geq \cdots \geq \sigma_{k-1} > \sigma_k$ and $1 + \frac{\eta_2}{k \mathcal{R}'_{\mathtt{ill}}(S)^2} > 0$, we know that $\sigma_{S'}^{\max} = \left( 1 + \frac{\eta_2}{k \mathcal{R}'_{\mathtt{ill}}(S)^2} \right) \sigma_1$ and $\sigma_{S'}^{\min} = \left( 1 + \frac{\eta_2}{k \mathcal{R}'_{\mathtt{ill}}(S)^2} - \frac{\eta_2}{\mathcal{R}'_{\mathtt{ill}}(S)^2} \right) \sigma_k$. Finally,

$$
\begin{aligned}
\kappa(S') &= \frac{\sigma_{S'}^{\max}}{\sigma_{S'}^{\min}} \\
&= \frac{\left( 1 + \frac{\eta_2}{k \mathcal{R}'_{\mathtt{ill}}(S)^2} \right) \sigma_1}{\left( 1 + \frac{\eta_2}{k \mathcal{R}'_{\mathtt{ill}}(S)^2} - \frac{\eta_2}{\mathcal{R}'_{\mathtt{ill}}(S)^2} \right) \sigma_k} \\
&= \frac{1 + \frac{\eta_2}{k \mathcal{R}'_{\mathtt{ill}}(S)^2}}{1 + \frac{\eta_2}{k \mathcal{R}'_{\mathtt{ill}}(S)^2} - \frac{\eta_2}{\mathcal{R}'_{\mathtt{ill}}(S)^2}} \kappa(S) \\
&> \kappa(S).
\end{aligned}
$$

$\square$

## B.3. Proof of Theorem 4.2

**Theorem 4.2.** *For* $\boldsymbol{H}(\theta) = \theta^\top \boldsymbol{K}\theta$*, if its maximum and minimum singular values* $\sigma_1$ *and* $\sigma_k$ *are unique, then*

*(1)* $\nabla_\theta \mathcal{R}_{\mathtt{well}}(\boldsymbol{H}(\theta)) = 2\boldsymbol{K}\theta\left(\sigma_1 \boldsymbol{v}_1 \boldsymbol{v}_1^\top - \frac{1}{D_{\mathtt{in}}}\theta^\top \boldsymbol{K}\theta\right),$

*(2)* $\nabla_\theta \mathcal{R}_{\mathtt{ill}}(\boldsymbol{H}(\theta)) = \frac{2\boldsymbol{K}\theta\left(\sigma_k \boldsymbol{v}_k \boldsymbol{v}_k^\top - \frac{1}{k}\theta^\top \boldsymbol{K}\theta\right)}{\left(\frac{1}{2k}\|\theta^\top \boldsymbol{K}\theta\|_F^2 - \frac{1}{2}\sigma_k^2\right)^2}.$

*Proof.* Given that $\boldsymbol{H}(\theta) = \theta^\top \boldsymbol{K}\theta$ for $\boldsymbol{K} = \boldsymbol{X}^\top \boldsymbol{X}$, we know $\boldsymbol{H}(\theta)$ is symmetric and positive semidefinite. Therefore, for compact SVD $\boldsymbol{H}(\theta) = \boldsymbol{U}\mathrm{Diag}(\boldsymbol{\sigma})\boldsymbol{V}^\top$, we have $\boldsymbol{U} = \boldsymbol{V}$.

(1) When the maximum singular value $\sigma_1$ of $\boldsymbol{H}$ is unique, we know from Theorem A.1 (3) that $\mathcal{R}_{\mathtt{well}}(\boldsymbol{H}(\theta))$ is differentiable with respect to $\boldsymbol{H}$, and $\nabla_{\boldsymbol{H}}\mathcal{R}_{\mathtt{well}}(\boldsymbol{H}(\theta)) = \sigma_1 \boldsymbol{u}_1 \boldsymbol{v}_1^\top - \frac{1}{D_{\mathtt{in}}}\boldsymbol{H} = \sigma_1 \boldsymbol{v}_1 \boldsymbol{v}_1^\top - \frac{1}{D_{\mathtt{in}}}\boldsymbol{H}$.

Given the form $\boldsymbol{H}(\theta) = \theta^\top \boldsymbol{K}\theta$, we have $d\boldsymbol{H} = (d\theta)^\top \boldsymbol{K}\theta + \theta^\top \boldsymbol{K}(d\theta)$. Furthermore,

$$
\begin{aligned}
(d\mathcal{R}_{\mathtt{well}})(\boldsymbol{H}(\theta)) &= \langle \nabla_{\boldsymbol{H}}\mathcal{R}_{\mathtt{well}}(\boldsymbol{H}(\theta)), d\boldsymbol{H}\rangle_F \\
&= \mathrm{Tr}\left(\left(\sigma_1 \boldsymbol{v}_1 \boldsymbol{v}_1^\top - \frac{1}{D_{\mathtt{in}}}\boldsymbol{H}\right)^\top \left((d\theta)^\top \boldsymbol{K}\theta + \theta^\top \boldsymbol{K}(d\theta)\right)\right) \\
&= \mathrm{Tr}\left(\left(\sigma_1 \boldsymbol{v}_1 \boldsymbol{v}_1^\top - \frac{1}{D_{\mathtt{in}}}\boldsymbol{H}\right)^\top (d\theta)^\top \boldsymbol{K}\theta\right) + \mathrm{Tr}\left(\left(\sigma_1 \boldsymbol{v}_1 \boldsymbol{v}_1^\top - \frac{1}{D_{\mathtt{in}}}\boldsymbol{H}\right)^\top \theta^\top \boldsymbol{K}(d\theta)\right) \\
&= \mathrm{Tr}\left(\boldsymbol{K}\theta\left(\sigma_1 \boldsymbol{v}_1 \boldsymbol{v}_1^\top - \frac{1}{D_{\mathtt{in}}}\boldsymbol{H}\right)^\top (d\theta)^\top\right) + \mathrm{Tr}\left((d\theta)\left(\sigma_1 \boldsymbol{v}_1 \boldsymbol{v}_1^\top - \frac{1}{D_{\mathtt{in}}}\boldsymbol{H}\right)^\top \theta^\top \boldsymbol{K}\right),
\end{aligned}
$$

in which $\langle \cdot, \cdot \rangle_F$ denotes the Frobenius inner product, and that last equality follows from the cyclic property of trace. As a result, following the derivatives of traces as in Eq. (100) and Eq. (104) in Petersen et al. (2008),

$$
\begin{aligned}
\nabla_\theta \mathcal{R}_{\mathtt{well}}(\boldsymbol{H}(\theta)) &= \frac{\partial \mathcal{R}_{\mathtt{well}}(\boldsymbol{H}(\theta))}{\partial \theta} \\
&= \boldsymbol{K}\theta\left(\sigma_1 \boldsymbol{v}_1 \boldsymbol{v}_1^\top - \frac{1}{D_{\mathtt{in}}}\boldsymbol{H}\right)^\top + \left(\left(\sigma_1 \boldsymbol{v}_1 \boldsymbol{v}_1^\top - \frac{1}{D_{\mathtt{in}}}\boldsymbol{H}\right)^\top \theta^\top \boldsymbol{K}\right)^\top \\
&= \boldsymbol{K}\theta\left(\sigma_1 (\boldsymbol{v}_1 \boldsymbol{v}_1^\top)^\top - \frac{1}{D_{\mathtt{in}}}\boldsymbol{H}^\top\right) + \boldsymbol{K}^\top \theta\left(\sigma_1 \boldsymbol{v}_1 \boldsymbol{v}_1^\top - \frac{1}{D_{\mathtt{in}}}\boldsymbol{H}\right) \\
&= 2\boldsymbol{K}\theta\left(\sigma_1 \boldsymbol{v}_1 \boldsymbol{v}_1^\top - \frac{1}{D_{\mathtt{in}}}\boldsymbol{H}\right).
\end{aligned}
$$

(2) When the minimum singular value $\sigma_k$ of $\boldsymbol{H}$ is unique, we know from Theorem 4.1 (3) that $\mathcal{R}_{\mathtt{ill}}(\boldsymbol{H}(\theta))$ is differentiable with respect to $\boldsymbol{H}$, and $\nabla_{\boldsymbol{H}}\mathcal{R}_{\mathtt{ill}}(\boldsymbol{H}(\theta)) = \frac{\sigma_k \boldsymbol{u}_k \boldsymbol{v}_k^\top - \frac{1}{k}\boldsymbol{H}}{\left(\frac{1}{2k}\|\boldsymbol{H}\|_F^2 - \frac{1}{2}\sigma_k^2\right)^2}$. Following similar arguments as in (1), we have $\nabla_\theta \mathcal{R}_{\mathtt{ill}}(\boldsymbol{H}(\theta)) = \frac{2\boldsymbol{K}\theta\left(\sigma_k \boldsymbol{u}_k \boldsymbol{v}_k^\top - \frac{1}{k}\boldsymbol{H}\right)}{\left(\frac{1}{2k}\|\boldsymbol{H}\|_F^2 - \frac{1}{2}\sigma_k^2\right)^2}$.

$\square$

## B.4. Proof of Theorem 4.3

**Theorem 4.3.** *For the trainable feature extractor* $\theta$*, feature covariance* $\boldsymbol{H}_{\mathtt{P}}(\theta) = \theta^\top \boldsymbol{K}_{\mathtt{P}}\theta$ *of the primary task and* $\boldsymbol{H}_{\mathtt{H}}(\theta) = \theta^\top \boldsymbol{K}_{\mathtt{H}}\theta$ *of the immunization task with* $\mathrm{rank}(\boldsymbol{H}_{\mathtt{P}}) = k_{\mathtt{P}}$*,* $\mathrm{rank}(\boldsymbol{H}_{\mathtt{H}}) = k_{\mathtt{H}}$ *and compact SVD* $\boldsymbol{H}_{\mathtt{P}}(\theta) = \boldsymbol{U}_{\mathtt{P}}\mathrm{Diag}(\boldsymbol{\sigma}_{\mathtt{P}})\boldsymbol{V}_{\mathtt{P}}^\top$*,* $\boldsymbol{H}_{\mathtt{H}}(\theta) = \boldsymbol{U}_{\mathtt{H}}\mathrm{Diag}(\boldsymbol{\sigma}_{\mathtt{H}})\boldsymbol{V}_{\mathtt{H}}^\top$*, for* $\boldsymbol{\sigma}_{\mathtt{P}} = [\sigma_{\mathtt{P},1}, \cdots, \sigma_{\mathtt{P},k_{\mathtt{P}}}]$*,* $\boldsymbol{\sigma}_{\mathtt{H}} = [\sigma_{\mathtt{H},1}, \cdots, \sigma_{\mathtt{H},k_{\mathtt{H}}}]$*,*

*(1) if* $\sigma_{\boldsymbol{H}_{\mathtt{P}}}^{\max}$ *is unique, i.e.,* $\sigma_{\boldsymbol{H}_{\mathtt{P}}}^{\max} = \sigma_{\mathtt{P},1} > \sigma_{\mathtt{P},2}$*, update* $\theta$ *such that* $\theta' = \theta - \eta_{\mathtt{P}}\boldsymbol{K}_{\mathtt{P}}^{-1}\nabla_\theta \mathcal{R}_{\mathtt{well}}(\boldsymbol{H}_{\mathtt{P}}(\theta))$ *for* $0 < \eta_{\mathtt{P}} < \min\left\{\frac{1}{(1-\frac{1}{D_{\mathtt{in}}})\sigma_{\mathtt{P},1}}, \frac{\sqrt{\sigma_{\mathtt{P},1}\sigma_{\mathtt{P},2}} - \sigma_{\mathtt{P},2}}{\frac{2}{D_{\mathtt{in}}}\sigma_{\mathtt{P},2}^2}\right\}$*, then* $\kappa\left(\theta'^\top \boldsymbol{K}_{\mathtt{P}}\theta'\right) < \kappa\left(\theta^\top \boldsymbol{K}_{\mathtt{P}}\theta\right)$*,*

(2) *if $\sigma_{\boldsymbol{H}_{\mathrm{H}}}^{\min}$ is unique,* i.e., $\sigma_{\boldsymbol{H}_{\mathrm{H}}}^{\min} = \sigma_{\mathrm{H},k_{\mathrm{H}}} < \sigma_{\mathrm{H},k_{\mathrm{H}}-1}$, *update $\theta$ such that $\theta' = \theta - \eta_{\mathrm{H}} \boldsymbol{K}_{\mathrm{H}}^{-1} \nabla_{\theta} \mathcal{R}_{\mathrm{ill}}(\boldsymbol{H}_{\mathrm{H}}(\theta))$ for $0 < \eta_{\mathrm{H}} < \frac{1}{1-2\sigma_{\boldsymbol{H}_{\mathrm{H}}}^{\min}/k_{\mathrm{H}}} \left( \frac{1}{2k_{\mathrm{H}}} \left\| \theta^{\top} \boldsymbol{K}_{\mathrm{H}} \theta \right\|_F^2 - \frac{1}{2} \left( \sigma_{\boldsymbol{H}_{\mathrm{H}}}^{\min} \right)^2 \right)^2$, then $\kappa\left( \theta'^{\top} \boldsymbol{K}_{\mathrm{H}} \theta' \right) > \kappa\left( \theta^{\top} \boldsymbol{K}_{\mathrm{H}} \theta \right).$*

*Proof.* (1) By Theorem 4.2 (1), we know $\nabla_{\theta} \mathcal{R}_{\mathrm{well}}(\boldsymbol{H}_{\mathrm{P}}(\theta)) = 2\boldsymbol{K}_{\mathrm{P}} \theta \left( \sigma_{\mathrm{P},1} \boldsymbol{v}_{\mathrm{P},1} \boldsymbol{v}_{\mathrm{P},1}^{\top} - \frac{1}{D_{\mathrm{in}}} \boldsymbol{H}_{\mathrm{P}} \right)$. Since $\theta' = \theta - \eta_{\mathrm{P}} \boldsymbol{K}_{\mathrm{P}}^{-1} \nabla_{\theta} \mathcal{R}_{\mathrm{well}}(\boldsymbol{H}_{\mathrm{P}}(\theta))$, we have

$$
\begin{aligned}
\theta'^{\top} \boldsymbol{K}_{\mathrm{P}} \theta' &= \left( \theta - \eta_{\mathrm{P}} \boldsymbol{K}_{\mathrm{P}}^{-1} \nabla_{\theta} \mathcal{R}_{\mathrm{well}}(\boldsymbol{H}_{\mathrm{P}}(\theta)) \right)^{\top} \boldsymbol{K}_{\mathrm{P}} \left( \theta - \eta_{\mathrm{P}} \boldsymbol{K}_{\mathrm{P}}^{-1} \nabla_{\theta} \mathcal{R}_{\mathrm{well}}(\boldsymbol{H}_{\mathrm{P}}(\theta)) \right) \\
&= \left( \theta - 2\eta_{\mathrm{P}} \boldsymbol{K}_{\mathrm{P}}^{-1} \boldsymbol{K}_{\mathrm{P}} \theta \left( \sigma_{\mathrm{P},1} \boldsymbol{v}_{\mathrm{P},1} \boldsymbol{v}_{\mathrm{P},1}^{\top} - \frac{1}{D_{\mathrm{in}}} \boldsymbol{H}_{\mathrm{P}} \right) \right)^{\top} \boldsymbol{K}_{\mathrm{P}} \left( \theta - 2\eta_{\mathrm{P}} \boldsymbol{K}_{\mathrm{P}}^{-1} \boldsymbol{K}_{\mathrm{P}} \theta \left( \sigma_{\mathrm{P},1} \boldsymbol{v}_{\mathrm{P},1} \boldsymbol{v}_{\mathrm{P},1}^{\top} - \frac{1}{D_{\mathrm{in}}} \boldsymbol{H}_{\mathrm{P}} \right) \right) \\
&= \left( \theta - 2\eta_{\mathrm{P}} \theta \left( \sigma_{\mathrm{P},1} \boldsymbol{v}_{\mathrm{P},1} \boldsymbol{v}_{\mathrm{P},1}^{\top} - \frac{1}{D_{\mathrm{in}}} \boldsymbol{H}_{\mathrm{P}} \right) \right)^{\top} \boldsymbol{K}_{\mathrm{P}} \left( \theta - 2\eta_{\mathrm{P}} \theta \left( \sigma_{\mathrm{P},1} \boldsymbol{v}_{\mathrm{P},1} \boldsymbol{v}_{\mathrm{P},1}^{\top} - \frac{1}{D_{\mathrm{in}}} \boldsymbol{H}_{\mathrm{P}} \right) \right) \\
&= \theta^{\top} \boldsymbol{K}_{\mathrm{P}} \theta - 2\eta_{\mathrm{P}} \left( \sigma_{\mathrm{P},1} \boldsymbol{v}_{\mathrm{P},1} \boldsymbol{v}_{\mathrm{P},1}^{\top} - \frac{1}{D_{\mathrm{in}}} \boldsymbol{H}_{\mathrm{P}} \right)^{\top} \theta^{\top} \boldsymbol{K}_{\mathrm{P}} \theta - 2\eta_{\mathrm{P}} \theta^{\top} \boldsymbol{K}_{\mathrm{P}} \theta \left( \sigma_{\mathrm{P},1} \boldsymbol{v}_{\mathrm{P},1} \boldsymbol{v}_{\mathrm{P},1} - \frac{1}{D_{\mathrm{in}}} \boldsymbol{H}_{\mathrm{P}} \right) \\
&\quad + 4\eta_{\mathrm{P}}^2 \left( \sigma_{\mathrm{P},1} \boldsymbol{v}_{\mathrm{P},1} \boldsymbol{v}_{\mathrm{P},1}^{\top} - \frac{1}{D_{\mathrm{in}}} \boldsymbol{H}_{\mathrm{P}} \right)^{\top} \theta^{\top} \boldsymbol{K}_{\mathrm{P}} \theta \left( \sigma_{\mathrm{P},1} \boldsymbol{v}_{\mathrm{P},1} \boldsymbol{v}_{\mathrm{P},1} - \frac{1}{D_{\mathrm{in}}} \boldsymbol{H}_{\mathrm{P}} \right) \\
&= \boldsymbol{H}_{\mathrm{P}} - 2\eta_{\mathrm{P}} \left( \sigma_{\mathrm{P},1} \boldsymbol{v}_{\mathrm{P},1} \boldsymbol{v}_{\mathrm{P},1}^{\top} - \frac{1}{D_{\mathrm{in}}} \boldsymbol{H}_{\mathrm{P}} \right)^{\top} \boldsymbol{H}_{\mathrm{P}} - 2\eta_{\mathrm{P}} \boldsymbol{H}_{\mathrm{P}} \left( \sigma_{\mathrm{P},1} \boldsymbol{v}_{\mathrm{P},1} \boldsymbol{v}_{\mathrm{P},1} - \frac{1}{D_{\mathrm{in}}} \boldsymbol{H}_{\mathrm{P}} \right) \\
&\quad + 4\eta_{\mathrm{P}}^2 \left( \sigma_{\mathrm{P},1} \boldsymbol{v}_{\mathrm{P},1} \boldsymbol{v}_{\mathrm{P},1}^{\top} - \frac{1}{D_{\mathrm{in}}} \boldsymbol{H}_{\mathrm{P}} \right)^{\top} \boldsymbol{H}_{\mathrm{P}} \left( \sigma_{\mathrm{P},1} \boldsymbol{v}_{\mathrm{P},1} \boldsymbol{v}_{\mathrm{P},1} - \frac{1}{D_{\mathrm{in}}} \boldsymbol{H}_{\mathrm{P}} \right).
\end{aligned}
$$

Since $\boldsymbol{H}_{\mathrm{P}}(\theta) = \theta^{\top} \boldsymbol{K}_{\mathrm{P}} \theta$ for $\boldsymbol{K}_{\mathrm{P}} = \boldsymbol{X}_{\mathrm{P}}^{\top} \boldsymbol{X}_{\mathrm{P}}$ is symmetric and positive semidefinite, we know for $\boldsymbol{H}_{\mathrm{P}}(\theta) = \boldsymbol{U}_{\mathrm{P}} \mathrm{Diag}(\boldsymbol{\sigma}_{\mathrm{P}}) \boldsymbol{V}_{\mathrm{P}}^{\top}$, it holds that $\boldsymbol{U}_{\mathrm{P}} = \boldsymbol{V}_{\mathrm{P}}$. Furthermore,

$$
\begin{aligned}
\sigma_{\mathrm{P},1} \boldsymbol{v}_{\mathrm{P},1} \boldsymbol{v}_{\mathrm{P},1}^{\top} - \frac{1}{D_{\mathrm{in}}} \boldsymbol{H}_{\mathrm{P}} &= \sigma_{\mathrm{P},1} \boldsymbol{v}_{\mathrm{P},1} \boldsymbol{v}_{\mathrm{P},1}^{\top} - \frac{1}{D_{\mathrm{in}}} \sum_{i=1}^{k_{\mathrm{P}}} \sigma_{\mathrm{P},i} \boldsymbol{u}_{\mathrm{P},i} \boldsymbol{v}_{\mathrm{P},i}^{\top} \\
&= \left( 1 - \frac{1}{D_{\mathrm{in}}} \right) \sigma_{\mathrm{P},1} \boldsymbol{u}_{\mathrm{P},1} \boldsymbol{v}_{\mathrm{P},1}^{\top} - \frac{1}{D_{\mathrm{in}}} \sum_{i=2}^{k_{\mathrm{P}}} \sigma_{\mathrm{P},i} \boldsymbol{u}_{\mathrm{P},i} \boldsymbol{v}_{\mathrm{P},i}^{\top} \\
&= \boldsymbol{U}_{\mathrm{P}} \mathrm{Diag}(\tilde{\boldsymbol{\sigma}}_{\mathrm{P}}) \boldsymbol{V}_{\mathrm{P}}^{\top} \\
&= \boldsymbol{V}_{\mathrm{P}} \mathrm{Diag}(\tilde{\boldsymbol{\sigma}}_{\mathrm{P}}) \boldsymbol{V}_{\mathrm{P}}^{\top}
\end{aligned}
$$

for $\mathrm{Diag}(\tilde{\boldsymbol{\sigma}}_{\mathrm{P}}) = \left[ \left( 1 - \frac{1}{D_{\mathrm{in}}} \right) \sigma_{\mathrm{P},1}, -\frac{1}{D_{\mathrm{in}}} \sigma_{\mathrm{P},2}, \cdots, -\frac{1}{D_{\mathrm{in}}} \sigma_{\mathrm{P},k_{\mathrm{P}}} \right]$. Therefore, plugging this and the SVD of $\boldsymbol{H}_{\mathrm{P}}$ back in,

$$
\begin{aligned}
\theta'^{\top} \boldsymbol{K}_{\mathrm{P}} \theta' &= \boldsymbol{V}_{\mathrm{P}} \mathrm{Diag}(\boldsymbol{\sigma}_{\mathrm{P}}) \boldsymbol{V}_{\mathrm{P}}^{\top} - 2\eta_{\mathrm{P}} \left( \boldsymbol{V}_{\mathrm{P}} \mathrm{Diag}(\tilde{\boldsymbol{\sigma}}_{\mathrm{P}}) \boldsymbol{V}_{\mathrm{P}}^{\top} \right)^{\top} \boldsymbol{V}_{\mathrm{P}} \mathrm{Diag}(\boldsymbol{\sigma}_{\mathrm{P}}) \boldsymbol{V}_{\mathrm{P}}^{\top} \\
&\quad - 2\eta_{\mathrm{P}} \left( \boldsymbol{V}_{\mathrm{P}} \mathrm{Diag}(\boldsymbol{\sigma}_{\mathrm{P}}) \boldsymbol{V}_{\mathrm{P}}^{\top} \right)^{\top} \boldsymbol{V}_{\mathrm{P}} \mathrm{Diag}(\tilde{\boldsymbol{\sigma}}_{\mathrm{P}}) \boldsymbol{V}_{\mathrm{P}}^{\top} \\
&\quad + 4\eta_{\mathrm{P}}^2 \left( \boldsymbol{V}_{\mathrm{P}} \mathrm{Diag}(\tilde{\boldsymbol{\sigma}}_{\mathrm{P}}) \boldsymbol{V}_{\mathrm{P}}^{\top} \right)^{\top} \boldsymbol{V}_{\mathrm{P}} \mathrm{Diag}(\boldsymbol{\sigma}_{\mathrm{P}}) \boldsymbol{V}_{\mathrm{P}}^{\top} \boldsymbol{V}_{\mathrm{P}} \mathrm{Diag}(\tilde{\boldsymbol{\sigma}}_{\mathrm{P}}) \boldsymbol{V}_{\mathrm{P}}^{\top} \\
&= \boldsymbol{V}_{\mathrm{P}} \mathrm{Diag}(\boldsymbol{\sigma}_{\mathrm{P}}) \boldsymbol{V}_{\mathrm{P}}^{\top} - 2\eta_{\mathrm{P}} \boldsymbol{V}_{\mathrm{P}} \mathrm{Diag}(\tilde{\boldsymbol{\sigma}}_{\mathrm{P}}) \mathrm{Diag}(\boldsymbol{\sigma}_{\mathrm{P}}) \boldsymbol{V}_{\mathrm{P}}^{\top} - 2\eta_{\mathrm{P}} \boldsymbol{V}_{\mathrm{P}} \mathrm{Diag}(\boldsymbol{\sigma}_{\mathrm{P}}) \mathrm{Diag}(\tilde{\boldsymbol{\sigma}}_{\mathrm{P}}) \boldsymbol{V}_{\mathrm{P}}^{\top} \\
&\quad + 4\eta_{\mathrm{P}}^2 \boldsymbol{V}_{\mathrm{P}} \mathrm{Diag}(\tilde{\boldsymbol{\sigma}}_{\mathrm{P}}) \mathrm{Diag}(\boldsymbol{\sigma}_{\mathrm{P}}) \mathrm{Diag}(\tilde{\boldsymbol{\sigma}}_{\mathrm{P}}) \boldsymbol{V}_{\mathrm{P}}^{\top} \\
&= \boldsymbol{V}_{\mathrm{P}} \mathrm{Diag}(\boldsymbol{\sigma}_{\mathrm{P}}) \boldsymbol{V}_{\mathrm{P}}^{\top} - 2\eta_{\mathrm{P}} \boldsymbol{V}_{\mathrm{P}} \mathrm{Diag}(\tilde{\boldsymbol{\sigma}}_{\mathrm{P}} \odot \boldsymbol{\sigma}_{\mathrm{P}}) \boldsymbol{V}_{\mathrm{P}}^{\top} - 2\eta_{\mathrm{P}} \boldsymbol{V}_{\mathrm{P}} \mathrm{Diag}(\boldsymbol{\sigma}_{\mathrm{P}} \odot \tilde{\boldsymbol{\sigma}}_{\mathrm{P}}) \boldsymbol{V}_{\mathrm{P}}^{\top} \\
&\quad + 4\eta_{\mathrm{P}}^2 \boldsymbol{V}_{\mathrm{P}} \mathrm{Diag}(\tilde{\boldsymbol{\sigma}}_{\mathrm{P}} \odot \boldsymbol{\sigma}_{\mathrm{P}} \odot \tilde{\boldsymbol{\sigma}}_{\mathrm{P}}) \boldsymbol{V}_{\mathrm{P}}^{\top} \\
&= \boldsymbol{V}_{\mathrm{P}} \mathrm{Diag}\left( \boldsymbol{\sigma}_{\mathrm{P}} - 4\eta_{\mathrm{P}} \tilde{\boldsymbol{\sigma}}_{\mathrm{P}} \odot \boldsymbol{\sigma}_{\mathrm{P}} + 4\eta_{\mathrm{P}}^2 \tilde{\boldsymbol{\sigma}}_{\mathrm{P}} \odot \boldsymbol{\sigma}_{\mathrm{P}} \odot \tilde{\boldsymbol{\sigma}}_{\mathrm{P}} \right) \boldsymbol{V}_{\mathrm{P}}^{\top} \\
&= \boldsymbol{V}_{\mathrm{P}} \mathrm{Diag}(\boldsymbol{\sigma}_{\mathrm{P}}') \boldsymbol{V}_{\mathrm{P}}^{\top},
\end{aligned}
$$

in which $\boldsymbol{\sigma}_{\mathrm{P}}' = \left[ \sigma_{\mathrm{P},1}', \cdots, \sigma_{\mathrm{P},k_{\mathrm{P}}}' \right]^{\top}$ for $\sigma_{\mathrm{P},i}' = \begin{cases} \sigma_{\mathrm{P},1} - 4\eta_{\mathrm{P}} \left( 1 - \frac{1}{D_{\mathrm{in}}} \right) \sigma_{\mathrm{P},1}^2 + 4\eta_{\mathrm{P}}^2 \left( 1 - \frac{1}{D_{\mathrm{in}}} \right)^2 \sigma_{\mathrm{P},1}^3 & \text{if } i = 1 \\ \sigma_{\mathrm{P},i} + \frac{4\eta_{\mathrm{P}}}{D_{\mathrm{in}}} \sigma_{\mathrm{P},i}^2 + \frac{4\eta_{\mathrm{P}}^2}{D_{\mathrm{in}}^2} \sigma_{\mathrm{P},i}^3 & \text{if } i > 1 \end{cases}$, $\odot$ de-

notes element-wise product and the second equality holds by the fact that $\boldsymbol{V}_{\mathrm{P}}$ is orthonormal, i.e., $\boldsymbol{V}_{\mathrm{P}}^{\top} \boldsymbol{V}_{\mathrm{P}} = \boldsymbol{I}$.

Since $\sigma_{\boldsymbol{H}_\text{P}}^{\max}$ is unique, we know that $\exists\, \alpha > 1$ such that $\sigma_{\text{P},1} = \alpha\sigma_{\text{P},2}$. Therefore,

$$
\begin{aligned}
\sigma'_{\text{P},2} &= \sigma_{\text{P},2} + \frac{4\eta_\text{P}}{D_\text{in}}\sigma_{\text{P},2}^2 + \frac{4\eta_\text{P}^2}{D_\text{in}^2}\sigma_{\text{P},2}^3 \\
&= \left(1 + \frac{4\eta_\text{P}}{D_\text{in}}\sigma_{\text{P},2} + \frac{4\eta_\text{P}^2}{D_\text{in}^2}\sigma_{\text{P},2}^2\right)\sigma_{\text{P},2} \\
&= \frac{1 + \frac{4\eta_\text{P}}{D_\text{in}}\sigma_{\text{P},2} + \frac{4\eta_\text{P}^2}{D_\text{in}^2}\sigma_{\text{P},2}^2}{\alpha}\sigma_{\text{P},1}.
\end{aligned}
$$

With $\eta_\text{P} < \frac{\sqrt{\sigma_{\text{P},1}\sigma_{\text{P},2}} - \sigma_{\text{P},2}}{\frac{2}{D_\text{in}}\sigma_{\text{P},2}^2}$, we have $1 + \frac{4\eta_\text{P}}{D_\text{in}}\sigma_{\text{P},2} + \frac{4\eta_\text{P}^2}{D_\text{in}^2}\sigma_{\text{P},2}^2 < 1 - 4\eta_\text{P}\left(1 - \frac{1}{D_\text{in}}\right)\sigma_{\text{P},1} + 4\eta_\text{P}^2\left(1 - \frac{1}{D_\text{in}}\right)^2\sigma_{\text{P},1}^2$. As a result,

$$
\frac{1 + \frac{4\eta_\text{P}}{D_\text{in}}\sigma_{\text{P},2} + \frac{4\eta_\text{P}^2}{D_\text{in}^2}\sigma_{\text{P},2}^2}{1 - 4\eta_\text{P}\left(1 - \frac{1}{D_\text{in}}\right)\sigma_{\text{P},1} + 4\eta_\text{P}^2\left(1 - \frac{1}{D_\text{in}}\right)^2\sigma_{\text{P},1}^2} < 1 < \alpha,
$$

that is,

$$
\frac{1 + \frac{4\eta_\text{P}}{D_\text{in}}\sigma_{\text{P},2} + \frac{4\eta_\text{P}^2}{D_\text{in}^2}\sigma_{\text{P},2}^2}{\alpha} < 1 - 4\eta_\text{P}\left(1 - \frac{1}{D_\text{in}}\right)\sigma_{\text{P},1} + 4\eta_\text{P}^2\left(1 - \frac{1}{D_\text{in}}\right)^2\sigma_{\text{P},1}^2.
$$

Plugging this result back in,

$$
\begin{aligned}
\sigma'_{\text{P},2} &= \frac{1 + \frac{4\eta_\text{P}}{D_\text{in}}\sigma_{\text{P},2} + \frac{4\eta_\text{P}^2}{D_\text{in}^2}\sigma_{\text{P},2}^2}{\alpha}\sigma_{\text{P},1} \\
&< \left(1 - 4\eta_\text{P}\left(1 - \frac{1}{D_\text{in}}\right)\sigma_{\text{P},1} + 4\eta_\text{P}^2\left(1 - \frac{1}{D_\text{in}}\right)^2\sigma_{\text{P},1}^2\right)\sigma_{\text{P},1} \\
&= \sigma'_{\text{P},1}.
\end{aligned}
$$

In addition, $\sigma'_{\text{P},2} = \sigma_{\text{P},2} + \frac{4\eta_\text{P}}{D_\text{in}}\sigma_{\text{P},2}^2 + \frac{4\eta_\text{P}^2}{D_\text{in}^2}\sigma_{\text{P},2}^3 \geq \sigma_{\text{P},i} + \frac{4\eta_\text{P}}{D_\text{in}}\sigma_{\text{P},i}^2 + \frac{4\eta_\text{P}^2}{D_\text{in}^2}\sigma_{\text{P},i}^3 = \sigma'_{\text{P},i}$ for $i = 3, \cdots, k_\text{P}$ since $\sigma_{\text{P},2} \geq \sigma_{\text{P},i}$ for $i = 3, \cdots, k_\text{P}$ by definition. Therefore, $\sigma'_{\text{P},1}$ remains to be the maximum singular value of $\theta'^\top \boldsymbol{K}_\text{P}\theta'$, and $\sigma'_{\text{P},k_\text{P}}$ the minimum. Finally,

$$
\begin{aligned}
\kappa\left(\theta'^\top \boldsymbol{K}_\text{P}\theta'\right) &= \frac{\sigma'_{\text{P},1}}{\sigma'_{\text{P},k_\text{P}}} \\
&= \frac{\sigma_{\text{P},1} - 4\eta_\text{P}\left(1 - \frac{1}{D_\text{in}}\right)\sigma_{\text{P},1}^2 + 4\eta_\text{P}^2\left(1 - \frac{1}{D_\text{in}}\right)^2\sigma_{\text{P},1}^3}{\sigma_{\text{P},k_\text{P}} + \frac{4\eta_\text{P}}{D_\text{in}}\sigma_{\text{P},k_\text{P}}^2 + \frac{4\eta_\text{P}^2}{D_\text{in}^2}\sigma_{\text{P},k_\text{P}}^3} \\
&< \frac{\sigma_{\text{P},1} - 4\eta_\text{P}\left(1 - \frac{1}{D_\text{in}}\right)\sigma_{\text{P},1}^2 + 4\eta_\text{P}^2\left(1 - \frac{1}{D_\text{in}}\right)^2\sigma_{\text{P},1}^3}{\sigma_{\text{P},k_\text{P}}} \\
&< \frac{\sigma_{\text{P},1}}{\sigma_{\text{P},k_\text{P}}} \\
&= \kappa\left(\theta^\top \boldsymbol{K}_\text{P}\theta\right)
\end{aligned}
$$

where the second inequality holds when $\eta_\text{P} < \frac{1}{\left(1 - \frac{1}{D_\text{in}}\right)\sigma_{\text{P},1}}$ which indicates that $-4\eta_\text{P}\left(1 - \frac{1}{D_\text{in}}\right)\sigma_{\text{P},1}^2 + 4\eta_\text{P}^2\left(1 - \frac{1}{D_\text{in}}\right)^2\sigma_{\text{P},1}^3 < 0$.

(2) Denote $\mathcal{R}'_{\mathtt{ill}}(\boldsymbol{H}_{\mathtt{H}}) = \frac{1}{2}\sigma^2_{\mathtt{H},k_{\mathtt{H}}} - \frac{1}{2k_{\mathtt{H}}}\|\boldsymbol{H}_{\mathtt{H}}\|^2_F$, then by Theorem 4.2 (2), we know $\nabla_\theta \mathcal{R}_{\mathtt{ill}}(\boldsymbol{H}_{\mathtt{H}}(\theta)) = \frac{2\boldsymbol{K}_{\mathtt{H}}\theta\left(\sigma_{\mathtt{H},k_{\mathtt{H}}}\boldsymbol{u}_{\mathtt{H},k_{\mathtt{H}}}\boldsymbol{v}^\top_{\mathtt{H},k_{\mathtt{H}}} - \frac{1}{k_{\mathtt{H}}}\boldsymbol{H}_{\mathtt{H}}\right)}{\mathcal{R}'_{\mathtt{ill}}(\boldsymbol{H}_{\mathtt{H}})^2}$. Since $\theta' = \theta - \eta_{\mathtt{H}}\boldsymbol{K}^{-1}_{\mathtt{H}}\nabla_\theta \mathcal{R}_{\mathtt{ill}}(\boldsymbol{H}_{\mathtt{H}}(\theta))$, we have

$$\theta'^\top \boldsymbol{K}_{\mathtt{H}}\theta' = \left(\theta - \eta_{\mathtt{H}}\boldsymbol{K}^{-1}_{\mathtt{H}}\nabla_\theta \mathcal{R}_{\mathtt{ill}}(\boldsymbol{H}_{\mathtt{H}}(\theta))\right)^\top \boldsymbol{K}_{\mathtt{H}}\left(\theta - \eta_{\mathtt{H}}\boldsymbol{K}^{-1}_{\mathtt{H}}\nabla_\theta \mathcal{R}_{\mathtt{ill}}(\boldsymbol{H}_{\mathtt{H}}(\theta))\right)$$

$$= \theta^\top \boldsymbol{K}_{\mathtt{H}}\theta - \frac{2\eta_{\mathtt{H}}}{\mathcal{R}'_{\mathtt{ill}}(\boldsymbol{H}_{\mathtt{H}})^2}\left(\sigma_{\mathtt{H},k_{\mathtt{H}}}\boldsymbol{u}_{\mathtt{H},k_{\mathtt{H}}}\boldsymbol{v}^\top_{\mathtt{H},k_{\mathtt{H}}} - \frac{1}{k_{\mathtt{H}}}\boldsymbol{H}_{\mathtt{H}}\right)^\top \theta^\top \boldsymbol{K}_{\mathtt{H}}\theta - \frac{2\eta_{\mathtt{H}}}{\mathcal{R}'_{\mathtt{ill}}(\boldsymbol{H}_{\mathtt{H}})^2}\theta^\top \boldsymbol{K}_{\mathtt{H}}\theta\left(\sigma_{\mathtt{H},k_{\mathtt{H}}}\boldsymbol{u}_{\mathtt{H},k_{\mathtt{H}}}\boldsymbol{v}^\top_{\mathtt{H},k_{\mathtt{H}}} - \frac{1}{k_{\mathtt{H}}}\boldsymbol{H}_{\mathtt{H}}\right)$$

$$+ \frac{4\eta^2_{\mathtt{H}}}{\mathcal{R}'_{\mathtt{ill}}(\boldsymbol{H}_{\mathtt{H}})^4}\left(\sigma_{\mathtt{H},k_{\mathtt{H}}}\boldsymbol{u}_{\mathtt{H},k_{\mathtt{H}}}\boldsymbol{v}^\top_{\mathtt{H},k_{\mathtt{H}}} - \frac{1}{k_{\mathtt{H}}}\boldsymbol{H}_{\mathtt{H}}\right)^\top \theta^\top \boldsymbol{K}_{\mathtt{H}}\theta\left(\sigma_{\mathtt{H},k_{\mathtt{H}}}\boldsymbol{u}_{\mathtt{H},k_{\mathtt{H}}}\boldsymbol{v}^\top_{\mathtt{H},k_{\mathtt{H}}} - \frac{1}{k_{\mathtt{H}}}\boldsymbol{H}_{\mathtt{H}}\right)$$

$$= \boldsymbol{H}_{\mathtt{H}} - \frac{2\eta_{\mathtt{H}}}{\mathcal{R}'_{\mathtt{ill}}(\boldsymbol{H}_{\mathtt{H}})^2}\left(\sigma_{\mathtt{H},k_{\mathtt{H}}}\boldsymbol{u}_{\mathtt{H},k_{\mathtt{H}}}\boldsymbol{v}^\top_{\mathtt{H},k_{\mathtt{H}}} - \frac{1}{k_{\mathtt{H}}}\boldsymbol{H}_{\mathtt{H}}\right)^\top \boldsymbol{H}_{\mathtt{H}} - \frac{2\eta_{\mathtt{H}}}{\mathcal{R}'_{\mathtt{ill}}(\boldsymbol{H}_{\mathtt{H}})^2}\boldsymbol{H}_{\mathtt{H}}\left(\sigma_{\mathtt{H},k_{\mathtt{H}}}\boldsymbol{u}_{\mathtt{H},k_{\mathtt{H}}}\boldsymbol{v}^\top_{\mathtt{H},k_{\mathtt{H}}} - \frac{1}{k_{\mathtt{H}}}\boldsymbol{H}_{\mathtt{H}}\right)$$

$$+ \frac{4\eta^2_{\mathtt{H}}}{\mathcal{R}'_{\mathtt{ill}}(\boldsymbol{H}_{\mathtt{H}})^4}\left(\sigma_{\mathtt{H},k_{\mathtt{H}}}\boldsymbol{u}_{\mathtt{H},k_{\mathtt{H}}}\boldsymbol{v}^\top_{\mathtt{H},k_{\mathtt{H}}} - \frac{1}{k_{\mathtt{H}}}\boldsymbol{H}_{\mathtt{H}}\right)^\top \boldsymbol{H}_{\mathtt{H}}\left(\sigma_{\mathtt{H},k_{\mathtt{H}}}\boldsymbol{u}_{\mathtt{H},k_{\mathtt{H}}}\boldsymbol{v}^\top_{\mathtt{H},k_{\mathtt{H}}} - \frac{1}{k_{\mathtt{H}}}\boldsymbol{H}_{\mathtt{H}}\right).$$

Since $\boldsymbol{H}_{\mathtt{P}}(\theta) = \theta^\top \boldsymbol{K}_{\mathtt{H}}\theta$ for $\boldsymbol{K}_{\mathtt{H}} = \boldsymbol{X}^\top_{\mathtt{H}}\boldsymbol{X}_{\mathtt{H}}$ is also symmetric and positive semidefinite, we know for $\boldsymbol{H}_{\mathtt{H}}(\theta) = \boldsymbol{U}_{\mathtt{H}}\mathrm{Diag}(\boldsymbol{\sigma}_{\mathtt{H}})\boldsymbol{V}^\top_{\mathtt{H}}$, it holds that $\boldsymbol{U}_{\mathtt{H}} = \boldsymbol{V}_{\mathtt{H}}$. Following similar arguments as in (1),

$$\sigma_{\mathtt{H},k_{\mathtt{H}}}\boldsymbol{u}_{\mathtt{H},k_{\mathtt{H}}}\boldsymbol{v}^\top_{\mathtt{H},k_{\mathtt{H}}} - \frac{1}{k_{\mathtt{H}}}\boldsymbol{H}_{\mathtt{H}} = -\frac{1}{k_{\mathtt{H}}}\sum_{i=1}^{k_{\mathtt{H}}-1}\sigma_{\mathtt{H},i}\boldsymbol{u}_{\mathtt{H},i}\boldsymbol{v}^\top_{\mathtt{H},i} + \left(1 - \frac{1}{k_{\mathtt{H}}}\right)\sigma_{\mathtt{H},k_{\mathtt{H}}}\boldsymbol{u}_{\mathtt{H},k_{\mathtt{H}}}\boldsymbol{v}^\top_{\mathtt{H},k_{\mathtt{H}}}$$

$$= \boldsymbol{V}_{\mathtt{H}}\mathrm{Diag}(\tilde{\boldsymbol{\sigma}}_{\mathtt{H}})\boldsymbol{V}^\top_{\mathtt{H}}$$

for $\mathrm{Diag}(\tilde{\boldsymbol{\sigma}}_{\mathtt{H}}) = \left[-\frac{1}{k_{\mathtt{H}}}\sigma_{\mathtt{H},1}, \cdots, -\frac{1}{k_{\mathtt{H}}}\sigma_{\mathtt{H},k_{\mathtt{H}}-1}, \left(1 - \frac{1}{k_{\mathtt{H}}}\right)\sigma_{\mathtt{H},k_{\mathtt{H}}}\right]$. Since $\boldsymbol{V}_{\mathtt{H}}$ is orthonormal, i.e., $\boldsymbol{V}^\top_{\mathtt{H}}\boldsymbol{V}_{\mathtt{H}} = \boldsymbol{I}$,

$$\theta'^\top \boldsymbol{K}_{\mathtt{H}}\theta' = \boldsymbol{V}_{\mathtt{H}}\mathrm{Diag}(\boldsymbol{\sigma}_{\mathtt{H}})\boldsymbol{V}^\top_{\mathtt{H}} - \frac{2\eta_{\mathtt{H}}}{\mathcal{R}'_{\mathtt{ill}}(\boldsymbol{H}_{\mathtt{H}})^2}\left(\boldsymbol{V}_{\mathtt{H}}\mathrm{Diag}(\tilde{\boldsymbol{\sigma}}_{\mathtt{H}})\boldsymbol{V}^\top_{\mathtt{H}}\right)^\top \boldsymbol{V}_{\mathtt{H}}\mathrm{Diag}(\boldsymbol{\sigma}_{\mathtt{H}})\boldsymbol{V}^\top_{\mathtt{H}}$$

$$- \frac{2\eta_{\mathtt{H}}}{\mathcal{R}'_{\mathtt{ill}}(\boldsymbol{H}_{\mathtt{H}})^2}\left(\boldsymbol{V}_{\mathtt{H}}\mathrm{Diag}(\boldsymbol{\sigma}_{\mathtt{H}})\boldsymbol{V}^\top_{\mathtt{H}}\right)^\top \boldsymbol{V}_{\mathtt{H}}\mathrm{Diag}(\tilde{\boldsymbol{\sigma}}_{\mathtt{P}})\boldsymbol{V}^\top_{\mathtt{P}}$$

$$+ \frac{4\eta^2_{\mathtt{H}}}{\mathcal{R}'_{\mathtt{ill}}(\boldsymbol{H}_{\mathtt{H}})^4}\left(\boldsymbol{V}_{\mathtt{H}}\mathrm{Diag}(\tilde{\boldsymbol{\sigma}}_{\mathtt{H}})\boldsymbol{V}^\top_{\mathtt{H}}\right)^\top \boldsymbol{V}_{\mathtt{H}}\mathrm{Diag}(\boldsymbol{\sigma}_{\mathtt{H}})\boldsymbol{V}^\top_{\mathtt{H}}\boldsymbol{V}_{\mathtt{H}}\mathrm{Diag}(\tilde{\boldsymbol{\sigma}}_{\mathtt{H}})\boldsymbol{V}^\top_{\mathtt{H}}$$

$$= \boldsymbol{V}_{\mathtt{H}}\mathrm{Diag}\left(\boldsymbol{\sigma}_{\mathtt{H}} - \frac{4\eta_{\mathtt{H}}}{\mathcal{R}'_{\mathtt{ill}}(\boldsymbol{H}_{\mathtt{H}})^2}\tilde{\boldsymbol{\sigma}}_{\mathtt{H}}\odot \boldsymbol{\sigma}_{\mathtt{H}} + \frac{4\eta^2_{\mathtt{H}}}{\mathcal{R}'_{\mathtt{ill}}(\boldsymbol{H}_{\mathtt{H}})^4}\tilde{\boldsymbol{\sigma}}_{\mathtt{H}}\odot \boldsymbol{\sigma}_{\mathtt{H}}\odot \tilde{\boldsymbol{\sigma}}_{\mathtt{H}}\right)\boldsymbol{V}^\top_{\mathtt{H}}$$

$$= \boldsymbol{V}_{\mathtt{H}}\mathrm{Diag}(\boldsymbol{\sigma}'_{\mathtt{H}})\boldsymbol{V}^\top_{\mathtt{H}},$$

for $\boldsymbol{\sigma}'_{\mathtt{H}} = \left[\sigma'_{\mathtt{H},1}, \cdots, \sigma'_{\mathtt{H},k_{\mathtt{H}}}\right]^\top$, $\sigma'_{\mathtt{H},i} = \begin{cases} \sigma_{\mathtt{H},i} + \frac{4\eta_{\mathtt{H}}}{k_{\mathtt{H}}\mathcal{R}'_{\mathtt{ill}}(\boldsymbol{H}_{\mathtt{H}})^2}\sigma^2_{\mathtt{H},i} + \frac{4\eta^2_{\mathtt{H}}}{k^2_{\mathtt{H}}\mathcal{R}'_{\mathtt{ill}}(\boldsymbol{H}_{\mathtt{H}})^4}\sigma^3_{\mathtt{H},i} & \text{if } i < k_{\mathtt{H}} \\ \sigma_{\mathtt{H},k_{\mathtt{H}}} - \frac{4\eta_{\mathtt{H}}}{\mathcal{R}'_{\mathtt{ill}}(\boldsymbol{H}_{\mathtt{H}})^2}\left(1 - \frac{1}{k_{\mathtt{H}}}\right)\sigma^2_{\mathtt{H},k_{\mathtt{H}}} + \frac{4\eta^2_{\mathtt{H}}}{\mathcal{R}'_{\mathtt{ill}}(\boldsymbol{H}_{\mathtt{H}})^4}\left(1 - \frac{1}{k_{\mathtt{H}}}\right)^2\sigma^3_{\mathtt{H},k_{\mathtt{H}}} & \text{if } i = k_{\mathtt{H}} \end{cases}$,

and $\odot$ denotes element-wise product.

Since $\sigma^{\min}_{\boldsymbol{H}_{\mathtt{H}}}$ is unique, we know that $\exists \beta \in (0,1)$ such that $\sigma_{\mathtt{H},k_{\mathtt{H}}} = \beta\sigma_{\mathtt{H},k_{\mathtt{H}}-1}$. Then we have

$$\sigma'_{\mathtt{H},k_{\mathtt{H}}} = \sigma_{\mathtt{H},k_{\mathtt{H}}} - \frac{4\eta_{\mathtt{H}}}{\mathcal{R}'_{\mathtt{ill}}(\boldsymbol{H}_{\mathtt{H}})^2}\left(1 - \frac{1}{k_{\mathtt{H}}}\right)\sigma^2_{\mathtt{H},k_{\mathtt{H}}} + \frac{4\eta^2_{\mathtt{H}}}{\mathcal{R}'_{\mathtt{ill}}(\boldsymbol{H}_{\mathtt{H}})^4}\left(1 - \frac{1}{k_{\mathtt{H}}}\right)^2\sigma^3_{\mathtt{H},k_{\mathtt{H}}}$$

$$= \left(1 - \frac{4\eta_{\mathtt{H}}}{\mathcal{R}'_{\mathtt{ill}}(\boldsymbol{H}_{\mathtt{H}})^2}\left(1 - \frac{1}{k_{\mathtt{H}}}\right)\sigma_{\mathtt{H},k_{\mathtt{H}}} + \frac{4\eta^2_{\mathtt{H}}}{\mathcal{R}'_{\mathtt{ill}}(\boldsymbol{H}_{\mathtt{H}})^4}\left(1 - \frac{1}{k_{\mathtt{H}}}\right)^2\sigma^2_{\mathtt{H},k_{\mathtt{H}}}\right)\sigma_{\mathtt{H},k_{\mathtt{H}}}$$

$$= \left(1 - \frac{4\eta_{\mathtt{H}}}{\mathcal{R}'_{\mathtt{ill}}(\boldsymbol{H}_{\mathtt{H}})^2}\left(1 - \frac{1}{k_{\mathtt{H}}}\right)\sigma_{\mathtt{H},k_{\mathtt{H}}} + \frac{4\eta^2_{\mathtt{H}}}{\mathcal{R}'_{\mathtt{ill}}(\boldsymbol{H}_{\mathtt{H}})^4}\left(1 - \frac{1}{k_{\mathtt{H}}}\right)^2\sigma^2_{\mathtt{H},k_{\mathtt{H}}}\right)\beta\sigma_{\mathtt{H},k_{\mathtt{H}}-1}$$

$$= \left(1 + \frac{4\eta_{\mathtt{H}}\sigma_{\mathtt{H},k_{\mathtt{H}}}}{k_{\mathtt{H}}\mathcal{R}'_{\mathtt{ill}}(\boldsymbol{H}_{\mathtt{H}})^2} + \frac{4\eta^2_{\mathtt{H}}\sigma^2_{\mathtt{H},k_{\mathtt{H}}}}{k^2_{\mathtt{H}}\mathcal{R}'_{\mathtt{ill}}(\boldsymbol{H}_{\mathtt{H}})^4} - \frac{4\eta_{\mathtt{H}}\sigma_{\mathtt{H},k_{\mathtt{H}}}}{\mathcal{R}'_{\mathtt{ill}}(\boldsymbol{H}_{\mathtt{H}})^2} + \frac{4\eta^2_{\mathtt{H}}\sigma^2_{\mathtt{H},k_{\mathtt{H}}}}{\mathcal{R}'_{\mathtt{ill}}(\boldsymbol{H}_{\mathtt{H}})^4} - \frac{8\eta^2_{\mathtt{H}}\sigma^2_{\mathtt{H},k_{\mathtt{H}}}}{k_{\mathtt{H}}\mathcal{R}'_{\mathtt{ill}}(\boldsymbol{H}_{\mathtt{H}})^4}\right)\beta\sigma_{\mathtt{H},k_{\mathtt{H}}-1}.$$

Letting $0 < \eta_{\text{H}} < \frac{\mathcal{R}'_{\text{ill}}(\boldsymbol{H}_{\text{H}})^2}{1-2\sigma_{\text{H},k_{\text{H}}}/k_{\text{H}}} = \frac{1}{1-2\sigma_{\boldsymbol{H}_{\text{H}}}^{\min}/k_{\text{H}}} \left( \frac{1}{2k_{\text{H}}} \left\| \theta^{\top} \boldsymbol{K}_{\text{H}} \theta \right\|_F^2 - \frac{1}{2} \left( \sigma_{\boldsymbol{H}_{\text{H}}}^{\min} \right)^2 \right)^2$, we have

$$-\frac{4\eta_{\text{H}}\sigma_{\text{H},k_{\text{H}}}}{\mathcal{R}'_{\text{ill}}\left(\boldsymbol{H}_{\text{H}}\right)^2} + \frac{4\eta_{\text{H}}^2\sigma_{\text{H},k_{\text{H}}}^2}{\mathcal{R}'_{\text{ill}}\left(\boldsymbol{H}_{\text{H}}\right)^4} - \frac{8\eta_{\text{H}}^2\sigma_{\text{H},k_{\text{H}}}^2}{k_{\text{H}}\mathcal{R}'_{\text{ill}}\left(\boldsymbol{H}_{\text{H}}\right)^4} < 0. \tag{21}$$

Also, $1 - \frac{4\eta_{\text{H}}}{\mathcal{R}'_{\text{ill}}(\boldsymbol{H}_{\text{H}})^2} \left(1 - \frac{1}{k_{\text{H}}}\right) \sigma_{\text{H},k_{\text{H}}} + \frac{4\eta_{\text{H}}^2}{\mathcal{R}'_{\text{ill}}(\boldsymbol{H}_{\text{H}})^4} \left(1 - \frac{1}{k_{\text{H}}}\right)^2 \sigma_{\text{H},k_{\text{H}}}^2 = \left(1 - \frac{2\eta_{\text{H}}}{\mathcal{R}'_{\text{ill}}(\boldsymbol{H}_{\text{H}})^2} \left(1 - \frac{1}{k_{\text{H}}}\right) \sigma_{\text{H},k_{\text{H}}}\right)^2 > 0$ for any $\eta_{\text{H}} > 0$. Given that $\sigma_{\text{H},k_{\text{H}}-1} > \sigma_{\text{H},k_{\text{H}}}$ and Eq. (21),

$$\beta < 1$$
$$< \frac{1 + \frac{4\eta_{\text{H}}}{k_{\text{H}}\mathcal{R}'_{\text{ill}}(\boldsymbol{H}_{\text{H}})^2}\sigma_{\text{H},k_{\text{H}}-1} + \frac{4\eta_{\text{H}}^2}{k_{\text{H}}^2\mathcal{R}'_{\text{ill}}(\boldsymbol{H}_{\text{H}})^4}\sigma_{\text{H},k_{\text{H}}-1}^2}{1 + \frac{4\eta_{\text{H}}}{k_{\text{H}}\mathcal{R}'_{\text{ill}}(\boldsymbol{H}_{\text{H}})^2}\sigma_{\text{H},k_{\text{H}}} + \frac{4\eta_{\text{H}}^2}{k_{\text{H}}^2\mathcal{R}'_{\text{ill}}(\boldsymbol{H}_{\text{H}})^4}\sigma_{\text{H},k_{\text{H}}}^2}$$
$$< \frac{1 + \frac{4\eta_{\text{H}}}{k_{\text{H}}\mathcal{R}'_{\text{ill}}(\boldsymbol{H}_{\text{H}})^2}\sigma_{\text{H},k_{\text{H}}-1} + \frac{4\eta_{\text{H}}^2}{k_{\text{H}}^2\mathcal{R}'_{\text{ill}}(\boldsymbol{H}_{\text{H}})^4}\sigma_{\text{H},k_{\text{H}}-1}^2}{1 + \frac{4\eta_{\text{H}}}{\mathcal{R}'_{\text{ill}}(\boldsymbol{H}_{\text{H}})^2}\sigma_{\text{H},k_{\text{H}}} + \frac{4\eta_{\text{H}}^2}{k_{\text{H}}^2\mathcal{R}'_{\text{ill}}(\boldsymbol{H}_{\text{H}})^4}\sigma_{\text{H},k_{\text{H}}}^2 - \frac{4\eta_{\text{H}}}{\mathcal{R}'_{\text{ill}}(\boldsymbol{H}_{\text{H}})^2}\sigma_{\text{H},k_{\text{H}}} + \frac{4\eta_{\text{H}}^2}{\mathcal{R}'_{\text{ill}}(\boldsymbol{H}_{\text{H}})^4}\sigma_{\text{H},k_{\text{H}}}^2 - \frac{8\eta_{\text{H}}^2}{k_{\text{H}}\mathcal{R}'_{\text{ill}}(\boldsymbol{H}_{\text{H}})^4}\sigma_{\text{H},k_{\text{H}}}^2},$$

indicating $\left(1 + \frac{4\eta_{\text{H}}}{k_{\text{H}}\mathcal{R}'_{\text{ill}}(\boldsymbol{H}_{\text{H}})^2}\sigma_{\text{H},k_{\text{H}}} + \frac{4\eta_{\text{H}}^2}{k_{\text{H}}^2\mathcal{R}'_{\text{ill}}(\boldsymbol{H}_{\text{H}})^4}\sigma_{\text{H},k_{\text{H}}}^2 - \frac{4\eta_{\text{H}}}{\mathcal{R}'_{\text{ill}}(\boldsymbol{H}_{\text{H}})^2}\sigma_{\text{H},k_{\text{H}}} + \frac{4\eta_{\text{H}}^2}{\mathcal{R}'_{\text{ill}}(\boldsymbol{H}_{\text{H}})^4}\sigma_{\text{H},k_{\text{H}}}^2 - \frac{8\eta_{\text{H}}^2}{k_{\text{H}}\mathcal{R}'_{\text{ill}}(\boldsymbol{H}_{\text{H}})^4}\sigma_{\text{H},k_{\text{H}}}^2\right)\beta < 1 + \frac{4\eta_{\text{H}}}{k_{\text{H}}\mathcal{R}'_{\text{ill}}(\boldsymbol{H}_{\text{H}})^2}\sigma_{\text{H},k_{\text{H}}-1} + \frac{4\eta_{\text{H}}^2}{k_{\text{H}}^2\mathcal{R}'_{\text{ill}}(\boldsymbol{H}_{\text{H}})^4}\sigma_{\text{H},k_{\text{H}}-1}^2$. Therefore,

$$\sigma'_{\text{H},k_{\text{H}}} = \left(1 + \frac{4\eta_{\text{H}}\sigma_{\text{H},k_{\text{H}}}}{k_{\text{H}}\mathcal{R}'_{\text{ill}}\left(\boldsymbol{H}_{\text{H}}\right)^2} + \frac{4\eta_{\text{H}}^2\sigma_{\text{H},k_{\text{H}}}^2}{k_{\text{H}}^2\mathcal{R}'_{\text{ill}}\left(\boldsymbol{H}_{\text{H}}\right)^4} - \frac{4\eta_{\text{H}}\sigma_{\text{H},k_{\text{H}}}}{\mathcal{R}'_{\text{ill}}\left(\boldsymbol{H}_{\text{H}}\right)^2} + \frac{4\eta_{\text{H}}^2\sigma_{\text{H},k_{\text{H}}}^2}{\mathcal{R}'_{\text{ill}}\left(\boldsymbol{H}_{\text{H}}\right)^4} - \frac{8\eta_{\text{H}}^2\sigma_{\text{H},k_{\text{H}}}^2}{k_{\text{H}}\mathcal{R}'_{\text{ill}}\left(\boldsymbol{H}_{\text{H}}\right)^4}\right)\beta\sigma_{\text{H},k_{\text{H}}-1}$$
$$< \left(1 + \frac{4\eta_{\text{H}}}{k_{\text{H}}\mathcal{R}'_{\text{ill}}\left(\boldsymbol{H}_{\text{H}}\right)^2}\sigma_{\text{H},k_{\text{H}}-1} + \frac{4\eta_{\text{H}}^2}{k_{\text{H}}^2\mathcal{R}'_{\text{ill}}\left(\boldsymbol{H}_{\text{H}}\right)^4}\sigma_{\text{H},k_{\text{H}}-1}^2\right)\sigma_{\text{H},k_{\text{H}}-1}$$
$$= \sigma'_{\text{H},k_{\text{H}}-1}.$$

In addition, $\sigma'_{\text{H},k_{\text{H}}-1} = \sigma_{\text{H},k_{\text{H}}-1} + \frac{4\eta_{\text{H}}}{k_{\text{H}}\mathcal{R}'_{\text{ill}}(\boldsymbol{H}_{\text{H}})^2}\sigma_{\text{H},k_{\text{H}}-1}^2 + \frac{4\eta_{\text{H}}^2}{k_{\text{H}}^2\mathcal{R}'_{\text{ill}}(\boldsymbol{H}_{\text{H}})^4}\sigma_{\text{H},k_{\text{H}}-1}^3 \leq \sigma_{\text{H},i} + \frac{4\eta_{\text{H}}}{k_{\text{H}}\mathcal{R}'_{\text{ill}}(\boldsymbol{H}_{\text{H}})^2}\sigma_{\text{H},i}^2 + \frac{4\eta_{\text{H}}^2}{k_{\text{H}}^2\mathcal{R}'_{\text{ill}}(\boldsymbol{H}_{\text{H}})^4}\sigma_{\text{H},i}^3 = \sigma'_{\text{H},i}$ for $i = 1, \cdots, k_{\text{H}} - 2$ since $\sigma_{\text{H},k_{\text{H}}-1} \leq \sigma_{\text{H},i}$ for $i = 2, \cdots, k_{\text{H}} - 1$ by definition. That is to say, $\sigma'_{\text{H},k_{\text{H}}}$ remains to be the minimum singular value of $\theta'^{\top}\boldsymbol{K}_{\text{H}}\theta'$, and $\sigma'_{\text{H},1}$ the maximum. Finally,

$$\kappa\left(\theta'^{\top}\boldsymbol{K}_{\text{H}}\theta'\right)$$
$$= \frac{\sigma'_{\text{H},1}}{\sigma'_{\text{H},k_{\text{H}}}}$$
$$= \frac{\left(1 + \frac{4\eta_{\text{H}}}{k_{\text{H}}\mathcal{R}'_{\text{ill}}(\boldsymbol{H}_{\text{H}})^2}\sigma_{\text{H},1} + \frac{4\eta_{\text{H}}^2}{k_{\text{H}}^2\mathcal{R}'_{\text{ill}}(\boldsymbol{H}_{\text{H}})^4}\sigma_{\text{H},1}^2\right)\sigma_{\text{H},1}}{\left(1 + \frac{4\eta_{\text{H}}}{k_{\text{H}}\mathcal{R}'_{\text{ill}}(\boldsymbol{H}_{\text{H}})^2}\sigma_{\text{H},k_{\text{H}}} + \frac{4\eta_{\text{H}}^2}{k_{\text{H}}^2\mathcal{R}'_{\text{ill}}(\boldsymbol{H}_{\text{H}})^4}\sigma_{\text{H},k_{\text{H}}}^2 - \frac{4\eta_{\text{H}}}{\mathcal{R}'_{\text{ill}}(\boldsymbol{H}_{\text{H}})^2}\sigma_{\text{H},k_{\text{H}}} + \frac{4\eta_{\text{H}}^2}{\mathcal{R}'_{\text{ill}}(\boldsymbol{H}_{\text{H}})^4}\sigma_{\text{H},k_{\text{H}}}^2 - \frac{8\eta_{\text{H}}^2}{k_{\text{H}}\mathcal{R}'_{\text{ill}}(\boldsymbol{H}_{\text{H}})^4}\sigma_{\text{H},k_{\text{H}}}^2\right)\sigma_{\text{H},k_{\text{H}}}}$$
$$> \frac{\left(1 + \frac{4\eta_{\text{H}}}{k_{\text{H}}\mathcal{R}'_{\text{ill}}(\boldsymbol{H}_{\text{H}})^2}\sigma_{\text{H},1} + \frac{4\eta_{\text{H}}^2}{k_{\text{H}}^2\mathcal{R}'_{\text{ill}}(\boldsymbol{H}_{\text{H}})^4}\sigma_{\text{H},1}^2\right)\sigma_{\text{H},1}}{\left(1 + \frac{4\eta_{\text{H}}}{k_{\text{H}}\mathcal{R}'_{\text{ill}}(\boldsymbol{H}_{\text{H}})^2}\sigma_{\text{H},k_{\text{H}}} + \frac{4\eta_{\text{H}}^2}{k_{\text{H}}^2\mathcal{R}'_{\text{ill}}(\boldsymbol{H}_{\text{H}})^4}\sigma_{\text{H},k_{\text{H}}}^2\right)\sigma_{\text{H},k_{\text{H}}}}$$
$$> \frac{\sigma_{\text{H},1}}{\sigma_{\text{H},k_{\text{H}}}}$$
$$= \kappa\left(\theta^{\top}\boldsymbol{K}_{\text{H}}\theta\right)$$

where the first inequality holds by Eq. (21) and the second by $\sigma_{\text{H},1} > \sigma_{\text{H},k_{\text{H}}}$.

$\square$

# C. Detailed Experiment Setup

## C.1. Datasets

Stanford Cars (Krause et al., 2013) contains 16,185 images of 196 car models and focuses on fine-grained image classification. Country211 (Radford et al., 2021) is a dataset used for country classification based on satellite images, comprising 211 country-level labels, each with 150 training images. This is a subset of the YFCC100M dataset (Thomee et al., 2016) providing user-generated photos and videos, used for domain adaptation evaluation.

## C.2. Immunization training details

We summarize the hyper-parameters of training for model immunization in Tab. 4. We choose $\lambda_\text{P}$ and $\lambda_\text{H}$ by balancing the gradient norm of $\mathcal{R}_\text{well}$ and $\mathcal{R}_\text{ill}$. Specifically, we obtain the scale of $\lambda_\text{P}$ and $\lambda_\text{H}$ first and search over multiples of $\{1, 2, 3, 5\}$. For linear models, we search over the set of $\{0.0005, 0.001, 0.005, 0.01\}$ and report the best result. For ImageNet we followed the default learning rate $\eta = 1 \times 10^{-5}$. The number of epochs is based on early stopping using RIR and the test accuracy. All experiments are conducted using float64 precision to ensure numerical stability and reduce potential inaccuracies in computations.

*Table 4.* Hyperparameters for immunization training.

| Dataset | Model | $\eta$ | $\lambda_\text{P}$ | $\lambda_\text{H}$ | Epochs | $\mathcal{L}$ |
|---|---|---|---|---|---|---|
| HousePrice | Linear | 0.005 | 100 | $1 \times 10^7$ | 1000 | Mean squared error |
| MNIST | Linear | 0.001 | 1 | $5 \times 10^7$ | 30 | Binary Cross-entropy (CE) |
| ImageNet vs. Stanford Cars | ResNet18 | $1 \times 10^{-5}$ | $5 \times 10^{-5}$ | $2 \times 10^6$ | 3 | Label-smoothing CE |
| ImageNet vs. Country211 | ResNet18 | $1 \times 10^{-5}$ | $1 \times 10^{-4}$ | $2 \times 10^6$ | 3 | Label-smoothing CE |
| ImageNet vs. Stanford Cars | ViT | $1 \times 10^{-5}$ | $3 \times 10^{-6}$ | $3 \times 10^8$ | 2 | Label-smoothing CE |
| ImageNet vs. Country211 | ViT | $1 \times 10^{-5}$ | $1 \times 10^{-6}$ | $1 \times 10^8$ | 2 | Label-smoothing CE |

**Details of immunizing linear models.** For the regression task, the linear feature extractor $\theta \in \mathbb{R}^{79 \times 79}$ is a randomly initialized dummy linear layer, as discussed in Sec. 4.4. We handle missing values in the tabular data by filling NaNs with 0. Categorical features are converted into numerical values using LabelEncoder. Finally, the features and labels are normalized using their respective mean and standard deviation. To create $\mathcal{D}_\text{P}$ and $\mathcal{D}_\text{H}$, we split the House prices dataset (Montoya & DataCanary, 2016) by the feature MSZoning. Specifically, all entries where MSZoning = 'RL' are assigned to $\mathcal{D}_\text{H}$, while the remaining entries form $\mathcal{D}_\text{P}$.

For the binary classification task on MNIST, the linear feature extractor $\theta \in \mathbb{R}^{784 \times 784}$ is also a randomly initialized dummy linear layer, and we construct a training dataset by selecting two specific target digits. The dataset is created using a custom BinaryDataset class, which filters the original MNIST dataset to include only the chosen digits and assigns new labels: one digit is mapped to label 0 and the other to label 1. To ensure balance in the dataset, we limit the number of samples for each digit to the smaller count between the two. For optimization, we use Adam (Kingma, 2014) with $\beta = (.9, 0.999)$ and $\epsilon = 1 \times 10^{-8}$ instead of the basic gradient descent in Alg. 1. For the linear model, we computed the Hessian inverse by solving a regularized least-squares system, where the Hessian is in the shape of $\mathbb{R}^{D_\text{in} \times D_\text{in}}$. Here $\mathcal{D}_\text{in} = 79$ for the regression task and $\mathcal{D}_\text{in} = 784$ for the image classification task.

**Details of immunizing non-linear models.** The pre-trained ResNet18 and ViT are loaded from Pytorch Image Models (Wightman, 2019) with the model name resnet18 and vit_base_patch16_clip_224. We also create the dataset with the built-in function create_dataset from Wightman (2019). The feature embedding sizes for ResNet18 and ViT are 512 and 768, respectively. To facilitate balanced training when dataset sizes differ, we implement a CombinedLoader, which pairs batches from two data loaders. The longer dataset dictates training duration, while the shorter dataset cycles continuously using itertools. The number of epochs reported in Tab. 4 corresponds to the epochs of $\mathcal{D}_\text{H}$, *i.e.*, the shorter loader.

For optimization, we use SGD with Nesterov momentum to optimize Eq. (11), setting an initial learning rate of $1 \times 10^{-5}$ with momentum 0.9. The trainable feature extractor parameters are optimized with zero weight decay, while the classifier parameters use a weight decay of $2 \times 10^{-5}$.

## C.3. Pseudo-code of the dummy layer

We provide the Pseudo-code for implementing the dummy layer in Fig. 4 below. The `DummyLinear` layer extends `torch.nn.Linear` and incorporates an optional preconditioning mechanism in the backward pass using the inverse feature covariance matrix. The `LinearFunction` class defines the forward and backward computations, where the forward pass applies a standard linear transformation $XW^\top + b$ and stores the input, weight, and bias for gradient computation. In the backward pass, the input gradient is computed normally, while the weight gradient is modified based on whether preconditioning is enabled (`use_precond=True`). If enabled, the weight gradient is adjusted by solving a *regularized least-squares system* using the inverse of the feature covariance matrix $X^\top X + \epsilon I$, improving numerical stability.

```python
class LinearFunction:
    @staticmethod
    def forward(ctx, input, weight, bias, lambda_reg, use_precond):
        # Save input tensors for backward
        ctx.save_for_backward(input, weight, bias)
        ctx.lambda_reg = lambda_reg
        ctx.use_precond = use_precond

        # Compute output
        output = input.mm(weight.t())
        if bias is not None:
            output += bias.unsqueeze(0).expand_as(output)
        return output

    @staticmethod
    def backward(ctx, grad_output):
        # Retrieve saved tensors
        input, weight, bias = ctx.saved_tensors
        lambda_reg = ctx.lambda_reg
        use_precond = ctx.use_precond

        # Initialize gradients
        grad_input = grad_weight = grad_bias = None

        if ctx.needs_input_grad[0]:
            grad_input = grad_output.mm(weight)

        if ctx.needs_input_grad[1]:
            base_grad_weight = grad_output.t().mm(input)
            if use_precond:
                XtX = input.t().mm(input)
                lambda_eye = lambda_reg * torch.eye(XtX.size(0), device=XtX.device)
                XtX_reg = XtX + lambda_eye
                grad_weight = torch.linalg.solve(XtX_reg, base_grad_weight)
            else:
                grad_weight = base_grad_weight

        if bias is not None and ctx.needs_input_grad[2]:
            grad_bias = grad_output.sum(0)

        return grad_input, grad_weight, grad_bias, None, None

class DummyLinear(nn.Linear):
    def forward(self, input, lambda_reg, use_precond):
        # Dynamically decide whether to use the covariance inversion as the preconditioner
        return LinearFunction.apply(input, self.weight, self.bias, lambda_reg, use_precond)
```

*Figure 4.* Dummy layer with selective inverse feature covariance matrix in backward function.

