# OpenReview forum: "Model Immunization from a Condition Number Perspective"
_ICML.cc/2025/Conference — ICML 2025 oral_

### Official Review · Reviewer_uVHQ · 2025-03-11

**Overall Recommendation:** 5

**Summary:**

This paper proposes to achieve model immunization, that is, pretraining a model that is hard to fine-tune on some harmful tasks while preserving the performance of other tasks, by maximizing the condition number of the corresponding harmful fine-tuning objective so that the convergence is slow and numerically unstable. Specifically, they propose a differentiable regularization that is proved to increase the condition number of the matrix regularized and further extend such property to the model immunization setting for the linear model. Experiments are conducted for both linear models as well as nonlinear neural networks with various architectures and the proposed method is shown to effectively increase the condition number on the harmful task objective while preserving that of the primary objective.

---

## update after rebuttal

After reading the authors' rebuttal, I feel comfortable recommending **strong accept** for this paper.

**Claims And Evidence:**

The proposed regularizer is supported with theoretical proof to show that it monotonically increases the condition number, and is numerically verified that when applied for model immunization, the condition number of the harmful task indeed increases. The inferior convergence of such task with large condition number is also verified in experiments.

**Essential References Not Discussed:**

None to the best of the reviewer’s knowledge.

**Experimental Designs Or Analyses:**

The experiments mostly align with the theory and is extended to nonlinear neural networks. The proposed evaluation metric relative immunization ratio is intuitive and reasonable.

**Methods And Evaluation Criteria:**

The condition number is a well-known factor of convergence speed and stability in both the classic optimization and deep learning literature. It’s reasonable to motivate and evaluate model immunization from the condition number perspective.

**Other Comments Or Suggestions:**

N/A

**Other Strengths And Weaknesses:**

Strengths:

1. The proposed framework of achieving model immunization by manipulating the condition number of the objective function corresponding to different tasks is very novel.
2. The proposed regularizers that increase or decrease the condition number are theoretically well-supported and technically solid.
3. The numerical experiments demonstrate superior performance compared to baseline methods for model immunization.

Weaknesses:

1. Manipulating the condition number of objective functions seems computationally expensive. Even though the proposed regularizers are differentiable alternatives to the condition number, they still involve the maximum or minimum singular value of the regularized matrix. Could the authors justify how the proposed method could be generalized to an even larger scale?

**Questions For Authors:**

The proposed model immunization framework involves a primary task and a harmful task. How would the model perform on tasks other than the primary task and the harmful task?

**Relation To Broader Scientific Literature:**

The paper proposed a new method to achieve model immunization, a recent proposed concept in the broader context of AI safety, and serves as an alternative to IMMA, a bilevel optimization method proposed together with the concept of model immunization, while achieving better performance in terms of preserving the performance of the primary task.

**Theoretical Claims:**

I have checked the proof of Theorem 4.2, 4.3, and briefly 4.4. The proof seems sound.

---

> ### Author Rebuttal · Authors · 2025-03-30
>
> We thank the reviewer for the thoughtful and encouraging feedback. We are glad the reviewer found the theoretical formulation sound, the proposed regularizers technically solid, and the empirical results convincing. We address the specific concerns and questions below.
>
> > Q12. Manipulating the condition number of objective functions seems computationally expensive. Could the authors justify how the proposed method could be generalized to an even larger scale?
>
> The computational cost depends on two main factors: the number of samples and the complexity of computing singular values. For the first, in large-scale problems, the Hessian of the full dataset can be approximated using only a minibatch. For the second, as the reviewer points out, the proposed method requires only the maximum or minimum singular value of the regularized matrix, which (particularly in high-dimensional settings) can be efficiently computed using techniques such as the Lanczos algorithm [1,2] or randomized SVD [3,4]. These approaches reduce the computational complexity from $\mathcal{O}(d^3)$ for a full SVD of a $d \times d$ matrix to $\mathcal{O}_k(d^2)$, with the rank-dependent factor absorbed into $\mathcal{O}_k$. This reduction is beneficial given the typically low-rank structure of Hessian matrices.
>
> [1] Cullum and Willoughby. Lanczos algorithms for large symmetric eigenvalue computations: Vol. I: Theory. Society for Industrial and Applied Mathematics, 2002.
> [2] Golub and Van Loan. Matrix computations. JHU press, 2013.
> [3] Halko et. al. "Finding structure with randomness: Probabilistic algorithms for constructing approximate matrix decompositions." SIAM review, 2011.
> [4] Tropp. "Randomized algorithms for matrix computations." 2020.
>
> > Q13. How would the model perform on tasks other than the primary task and the harmful task?
>
> Thanks for pointing out this interesting direction. We now additionally report the ratio between condition numbers with and without immunization, i.e., Eq. 15 (i) but for all digits, for a model with $D_{\tt P}$ = digit 0 and $D_{\tt H}$ = digit 1.
>
> For digits other than $D_{\tt P}$ and $D_{\tt H}$, we observe that the ratio remains close to 1, indicating that immunization does not affect the condition number of the features on other tasks.
>
> For a theoretical perspective, following the reasoning in Sec. 3.1, the performance on other tasks is intuitively related to the correlation with $D_{\tt H}$, i.e., the relative angle between the singular vectors.
>
>
> |             | 0 ($D_{\tt P}$) | 1 ($D_{\tt H}$) | 2    | 3    | 4    | 5    | 6    | 7    | 8    | 9    |
> |--------------|----------------|-----------------|------|------|------|------|------|------|------|------|
> | condition number ratio | 0.0451         | 9.8571          | 1.7440 | 1.2147 | 0.9582 | 1.0049 | 0.9785 | 1.3007 | 2.2467 | 1.1453 |

---

> > ### Comment · Reviewer_uVHQ · 2025-04-04
> >
> > Thank you for your response! The rebuttal has already addressed my concerns and I am happy to recommend **accept** for this paper.

---

### Official Review · Reviewer_CAgr · 2025-03-12

**Overall Recommendation:** 5

**Summary:**

The paper reframes the immunization task, i.e., make models robust against finetuning on specific task, (Contribution 1: Section 3) from a novel perspective using condition number.

Through this insight, the authors propose a regularization method at the pretraining stage that makes finetuning for specific tasks more difficult (Contribution 2: Section 4.1, 4.2). They theoretically prove that using this regularization increases the condition number for tasks they aim to learn and decreases it for tasks they want to immunize against in linear model setting (Contribution 3: Section 4.3). To demonstrate the effectiveness of this regularization, they empirically showed improved relative immunization ratio for linear regression and image classification tasks on both linear models (Contribution 4-1: Section 5.1) and deep neural networks (Contribution 4-2: Section 5.2).

**Claims And Evidence:**

Theoretical claims and evidence: The main framework of condition number and its theoretical connection to the immunization task is very novel and clear.

Experimental evidence: This will be discussed in detail in the methods and evaluation criteria section.

**Essential References Not Discussed:**

Adversarial prompt to generate unsafe images by T2I models could be one of problem that can be solved from an immunization perspective. Please consider add this kind of technique.

[1]: Circumventing Concept Erasure Methods For Text-to-Image Generative Models, https://arxiv.org/abs/2308.01508
[2]: Ring-A-Bell! How Reliable are Concept Removal Methods for Diffusion Models?, https://arxiv.org/abs/2310.10012

**Experimental Designs Or Analyses:**

As stated in Methods And Evaluation Criteria, not addressing generative models and using only RIR as a performance metric for immunization are major issues.

**Methods And Evaluation Criteria:**

As the paper itself acknowledges (In section 4.4), its biggest weakness is the experimental setting. I would like to see two major improvements from the rebuttal.

First, there are no results for generative models. As mentioned in the introduction regarding text-to-image models, immunization is a more important issue for generation than classification. While the theoretical contribution of this paper is sufficiently commendable, it's difficult to strongly recommend it due to the lack of coverage on generative models.

For the immunization on generative model, I would be satisfied only on the simple dataset like MNIST. For example, it would be good to compare convergence speed after training an unconditional or conditional diffusion model on data excluding the digit 7 and then finetuning on 7. But if you have a preferred setting, feel free to use that.

Second, the paper heavily relies on the RIR metric. While I agree that immunization can be approximated by RIR, due to the limitations of 2nd order approximation of loss landscapes in neural networks (unlike linear models), it's difficult to claim that improvements in RIR directly translate to improvements in resistance on actual finetuning. For deep neural networks, I would like to see **test accuracy** on both D_H and D_p simultaneously. Drawing a Pareto curve w.r.t finetuning epochs or hyperparameter and showing superior results than baseline methods would be ideal. Currently, immunization on D_H is only reported through RIR, which I think doesn't accurately show how effective this method actually is for DNNs.

**The second problem appears quite critical to me**. I find it difficult to improve my score to "accept" if this experiment is not conducted. (Minor Q) Are there any papers dealing with immunization in non-linear settings that use RIR as the main evaluation metric? If so, please share them and I will take that into consideration.

---

If both issues are **perfectly** addressed, I think this paper has **oral-level** contribution.

**Other Comments Or Suggestions:**

(This is only a suggestion, so I would appreciate it if the authors could consider them and incorporate them at their favor.)

S1: Is the proof sketch of theorem 4.1 really necessary for the main text? I believe that even if the paper is aimed at a theoretical point, this part should be relegated to the appendix and more experimental results should be presented.

**Other Strengths And Weaknesses:**

The writing is clear, the motivation is also very good. I think the only weaknesses of this paper is the experiment setting.

**Questions For Authors:**

Q1: I'm curious about numerical stability of regularization term due to the 1/x form. What are the conditions for gradient explosion? Looking at eq (13), it seems like explosion occurs when S becomes a diagonal matrix composed of sigma^min, is that correct? I'm curious if this is the only case where explosion occurs. If so, since Hessian matrices typically have low rank structure, numerical stability could be further justified.

This question is just to improve the theoretical contribution of this paper.

**Relation To Broader Scientific Literature:**

As far as I know, this research is the first to apply condition number to a field other than optimization contexts related to convergence speed. Condition number seems to have potential applications in areas beyond safety (the goal of this research), such as model interpretability and other fields. I would like to give high credit for being the first research to apply condition number to a different field.

**Theoretical Claims:**

I was not able to verify all proofs provided in the appendix. The theorem results seem reasonable.

---

> ### Author Rebuttal · Authors · 2025-03-30
>
> We thank the reviewer for the thoughtful review and respond to questions below.
>
> > Q5. Generative task
>
> We consider the immunization against linear probing (Sec. 3). For generative tasks, linear probing is not commonly used for transfer learning. Instead, other techniques, e.g., LoRA, are more common. Nevertheless, we now report on immunization against linear probing for a generative model.
>
> We experiment with CVAE on the MNIST dataset. Let $D_{\tt P}$ be digit 0 images and $D_{\tt H}$ be digit 1 images. The CVAE uses MLP for both the encoder and decoder. The last layer of the decoder is used for linear probing, and all other layers are treated as the feature extractor. The loss $\mathcal{L}$ is the negative ELBO.
>
> As in the paper, we report RIR to evaluate immunization. To evaluate generation quality, we report the accuracy of a digit classifier (with 99.3% acc. on the MNIST test set).
>
> In **Tab R1**, our approach successfully immunizes the model against $D_{\tt H}$ without hurting the performance on $D_{\tt P}$, as indicated by a large RIR and acc. of 100% on the generated images.
>
> In **Tab R2**, we report linear probing on $D_{\tt H}$ across different epochs on models w/ and w/o immunization. We observe that the acc. of the model w/o immunization gradually learns to generate $D_{\tt H}$ (digit 1), while the immunized model struggles to learn $D_{\tt H}$ (digit 1).
>
> **Tab R1.** Results of immunizing CVAE on MNIST.
> || Metrics|
> |-|-|
> |Eq. 17 (i)|1128.4 |
> |Eq. 17 (ii)| 0.016|
> |RIR|68971.8|
> |Classification acc. on $D_{\tt P}$  |1.0 |
>
> **Tab R2.** Acc. of generation on $D_{\tt H}$ w/ and w/o immunization.
> |Linear probing epoch|5|10|15|20|25|
> |-|-|-|-|-|-|
> |w/o Immu. |0.30|0.37|0.47|0.54|0.69|
> |w/ Immu. |0.01|0.00|0.02|0.01|0.01|
>
> > Q6. Test acc. on $D_{\tt H}$ for deep nets v.s. fine-tuning epochs
>
> The acc. on $D_{\tt P}$ is fixed as reported in Tab.3 in our paper. In Tab. R3 & R4, we further report the linear probed (fine-tuned) results on different feature extractors and provide the test acc. on $D_{\tt H}$ during the first 20 epochs of linear probing (fine-tuning). Here, $D_{\tt H}$ refers to the Stanford Cars dataset. The acc. are reported every 5 epochs. As shown, our method exhibits the slowest convergence rate with on both ResNet18 and ViT; indicated by the lowest acc. compared with baselines. We will include a plot visualizing these results in the paper.
>
> **Tab R3.** Test acc. on $D_{\tt H}$ with ResNet18 as the backbone.
> |Method/Fine-tuning epoch|5|10|15|20|
> |-|-|-|-|-|
> |Pre-trained model|13.6|18.2|21.0|23.8|
> |$\mathcal{R}_{\tt ill}$ Only | 12.8 | 16.6 | 19.9 | 23.8 |
> |IMMA|16.8|20.9|23.7|26.4|
> |Opt $\kappa$|12.9|17.8|19.9|23.3|
> |$\bf Ours$|9.5|15.5|18.0|21.4|
>
> **Tab R4.** Test acc. on $D_{\tt H}$ with ViT as the backbone.
> |Method/Fine-tuning epoch|5|10|15|20|
> |-|-|-|-|-|
> |Pre-trained model|30.7|42.2|51.4|60.3|
> |$\mathcal{R}_{\tt ill}$ Only|8.8|20.9|23.3|39.0|
> |IMMA|23.31|35.7|47.4|58.7|
> |Opt $\kappa$|11.8|20.1|27.5|42.1|
> |$\bf Ours$|7.9|14.6|24.5|34.0|
>
> > Q7. Any other papers use RIR for evaluation?
>
> To the best of our knowledge, we are the first to study immunization from a condition number perspective. Therefore, we are not aware of other works using RIR as an evaluation metric.
>
> > Q8. Should the dummy layer be applied to every weight parameter in deep nets?
>
> Under the linear probing setting, the dummy layer only needs to be inserted at the last layer of the feature extractor, and the rest of the layers can be trained normally. Intuitively, we are treating a deep-net to be extracting "learnable feature" $\mathbf{x}$ followed by a linear feature extractor $\theta_L$, that is, we view a deep-net $f_{[\theta_1, \dots \theta_L]} = \mathbf{x}(\theta_1,\dots, \theta_{L-1})^\top\theta_{L}$. Note, our theoretical result is limited to linear models and does not fully justify such constructions. We believe this is an important future direction, and we believe the linear model's results provide a promising first step.
>
> > Q9. References on adversarial prompting
>
> Thanks! We will review the suggested [1, 2] along with other related works on concept erasing methods to provide a more comprehensive view of the field.
>
> > Q10. Suggestion on proof sketch
>
> We will adjust the length/placement of the proof sketches. This would also help to create space for the additionally suggested experiments for the main text.
>
> > Q11. Numerical stability of Reg. terms
>
> Indeed, $\mathbf{S}$ being a diagonal matrix composed of $\sigma_\min$'s is the only case for gradient explosion. We would note, though, that the premise of our theorems, e.g., Eq. (13), is the minimum singular value $\sigma_\min$ being unique, which would prevent this case in theoretical derivation. In practice, even though the Hessian could be low-rank, we observed that $\sigma_\max$ is usually much larger than $\sigma_\min$, in many cases to the extent of several orders of magnitude. Therefore, we did not empirically observe issues with gradient explosion.

---

> > ### Comment · Reviewer_CAgr · 2025-04-04
> >
> > I thank the author for their efforts. Almost all of my questions and concerns have been resolved, and I think this is an excellent paper that proposes a novel and principled perspective on the immunization / safety / unlearning research. There is still a severe limitation in that it can only be applied to linear probing, but nonetheless, I think it is excellent research that applies optimization theory to safety. I am upgrading my score to **strong accept**.
> >
> > > Q5. Generative task
> >
> > First of all, thank you for experimenting with generative models, which you could have considered out of scope.
> >
> > The results on CVAE look promising. The experiments on CVAE can be seen as a proof of concept that this methodology can also be applied to generative models. Since it has been partly shown that the impact of this research can be applied to generative models, I would like to adjust my score upward.
> >
> > However, one thing I'd like to mention is that when writing my review, I was considering applying the condition number to the entire set of parameters (or lora parameter) rather than just linear probing for generative models. Based on the insights from this research, I hope that better and more efficient regularization methods that can be applied to more complex models beyond linear models will emerge in subsequent studies.
> >
> > > Q6. Test acc. on $D_H$
> >
> > First, thank you for showing the test accuracy on $D_H$ w.r.t. epoch. I thought this result was **very important** in evaluating this paper. I think Tab R3 is the one which verifies that regularization through the conditional number has a positive effect on immunization.
> >
> > Personally, I think the RIR metric is closely related to what this method directly optimizes. Therefore, I am skeptical about using RIR as the only and primary metric for immunization (in neural network setting). In real-world scenario, what people are interested in when immunizing models is not the RIR but the actual evolution of test accuracy w.r.t. finetuning epochs. I'm curious about the author's thoughts as well.
> >
> > > Q7. Any other papers use RIR for evaluation?
> >
> > I think not using RIR corresponds to a proxy for immunization, not immunization itself. I find it difficult to agree with using a proxy as the main metric when evaluating immunization is computationally feasible.
> >
> > > Q8, Q9, Q10
> >
> > My confusions have been resolved. Thank you for accepting my suggestions and opinions.
> >
> > > Q11. Numerical stability of Reg. terms
> >
> > Correct. My intention wasn't to criticize that it might not be stable, but to say that it being stable is very reasonable. Since readers interested in numerical stability may be concerned due to the form of regularization, it might be good to include a brief discussion in the appendix or main manuscript (just a line or two) mentioning that the Hessian generally has $\sigma_max$ >> $\sigma_min$ [1]. I'm sharing a related reference: [1]
> >
> > [1]: Gradient Descent Happens in a Tiny Subspace, https://arxiv.org/abs/1812.04754

---

### Official Review · Reviewer_c1aR · 2025-03-13

**Overall Recommendation:** 4

**Summary:**

This paper proposed a framework for studying model immunization, i.e., the task to make fine-tuning on harmful datasets harder. The authors proposed that for linear models, the difficulty of fine-tuning can be characterized by the condition number of the Hessian matrix. Based on this theory, the authors proposed two regularizers to increasing the condition number of harmful dataset while keeping the condition number stable for the pre-training task. The authors further proved that both regularizers are differentiable, and the optimization goal can be achieved through gradient-based algorithm. Empirical result on linear models supports the theory and shows better performance than other baselines. The authors also empirically evaluated the method on two non-linear models ResNet18 and ViT. The results are also promising and better than baselines.

## update after rebuttal

Thank you for the further clarifications from rebuttal. They have addressed all my concerns. I personally agree immunization is an interesting and important direction to explore more.

**Claims And Evidence:**

* Claim: immunization to harmful fine-tuning, at least for linear models, can be charactered using the condition number of the Hessian matrix.
* Evidence: theoretically, condition number is a known indicator for the difficulty of gradient-based optimization; empirically, the experiment results, including the proposed method and $R_{ill}$-based approach also support the claim.

* Claim: (linear) model immunization can be modeled as an optimization problem with the objective shown in equation (11), and can be solved with algorithm 1.
* Evidence: (1) minimizing $R_{ill}$ will increase the condition number of harmful dataset, this regularizer is proven to be optimizable using gradient descent with guaranteed increase in condition number. (2) the regularizer $R_{well}$ is proven by Nenov et al. to be optimizable using gradient descent. (3) both regularizers are proven to be differentiable w.r.t. model parameters $\theta$, and has closed-form gradients. (4) an implementation of algorithm 1 in PyTorch and successful evaluation.

* Claim: solving equation (11) would make the model hard to fine-tune on harmful datasets while maintain a good pre-training task performance.
* Evidence: empirical experimental results on linear regression and image classificationl

* Claim: solving equation (11) also shows promising results on non-linear models.
* Evidence: empirical experimental results on ResNet18 and ViT.

**Essential References Not Discussed:**

Have not found.

**Experimental Designs Or Analyses:**

The experiments use three baselines for comparison. The selection of these baselines are reasonable.

The experiments use a few exemplary tasks for linear and non-linear models. While the tasks and datasets are all common and widely used, the selection of these tasks and corresponding dataset is not fully justified. Therefore, it's possible that the proposed method would not work as well on other tasks and datasets, especially for non-linear models.

For ResNet18, the last two convolutional blocks are updated; for ViT, the final transformer block is updated. Not sure why this setup.

**Methods And Evaluation Criteria:**

The evaluation criteria, relative immunization ratio (RIR), make sense. The paper also provided condition numbers for both harmful dataset and pre-training dataset.

**Other Comments Or Suggestions:**

No

**Other Strengths And Weaknesses:**

Maybe it would be obvious for experts, but I would hope to see some explanations for the challenges for modeling immunization for non-linear models.

**Questions For Authors:**

* What are the challenges for modeling immunization for non-linear models?
* Can the proposed framework generalize to non-linear models?

**Relation To Broader Scientific Literature:**

Model immunization is a promising approach to make open-weight models safer against malicious fine-tuning. This paper advances the state-of-the-art in this field.

**Theoretical Claims:**

Seems correct, have not checked carefully.

---

> ### Author Rebuttal · Authors · 2025-03-30
>
> We thank the reviewer for the positive and constructive review. We answer individual questions below.
>
> > Q1. Justification for task and dataset selection
>
> Our experiments consist of two settings: (a) a linear model setup, where the setting matches our proposed theoretical framework, and (b) a deep-net setup, where we experimented with non-linear models despite the theoretical gap.
>
> **For linear models**: For the regression task, we choose the House Price dataset as it is a widely used tabular dataset, e.g., in intro ML courses. For the classification task, we choose MNIST as it is the most basic image classification dataset. Additionally, linear models are effective on these datasets.
>
> **For deep-nets**: The transfer learning setting follows from [A] (See Line 352). The chosen Stanford Cars dataset and Country211 dataset are simply the first two datasets presented in [A]'s Figure. 4 (the official ICML version), where they demonstrate linear probing to work well. We now provide experimental results on the third dataset from their Figure. 4 (Food-101) as $D_{\tt H}$ in the table below. Here the feature extractor backbone is ResNet18. We observe that the proposed method is also effective in Food-101.
>
> - [A] Radford, Alec, et al. "Learning transferable visual models from natural language supervision." ICML 2021.
>
> | $D_{\tt H}$ | Eq. 17 (i) $\uparrow$ | Eq. 17 (ii) $\downarrow$ | $\tt RIR_{\theta_0}$ $\uparrow$| $D_{\tt P}$ Test Acc. $\uparrow$ |
> | -------- | -------- | -------- |-------- |-------- |
> | Food-101   |  1.712 |  0.571   | 3.045 |  63.74% |
>
> We will clarify the motivation of these setups in the experiment section.
>
>
>
> > Q2. Explanation of the experiment setup for ResNet18 and ViT
>
> We choose to update only the last two convolutional blocks of ResNet18 and the final transformer block of the Vision Transformer (ViT) following common practices in transfer learning. For this experimental setup, we are starting the immunization with a model pre-trained on ImageNet. Typically, only the final layers are updated for efficiency reasons. We will clarify this choice.
>
> We now provide the results of immunizing the entire ResNet18 backbone on the Stanford Cars dataset below. When training the entire backbone, we observe a similar result, but with a slightly lower immunization effect on $D_{\tt H}$ and test accuracy on $D_{\tt P}$. Note that the running time for the full model is more than twice that of only updating the last blocks.
>
> | Trainable module | Eq. 17 (i) $\uparrow$ | Eq. 17 (ii) $\downarrow$ | $\tt RIR_{\theta_0}$ $\uparrow$| $D_{\tt P}$ Test Acc. $\uparrow$ |
> | -------- | -------- | -------- |-------- |-------- |
> | Entire ResNet18    | 2.102   | 0.672    | 3.127  | 61.48%   |
> | Last two blocks    | 2.386    |0.699  |  3.467   | 62.36% |
>
>
>
> > Q3. The challenges for modeling immunization for non-linear models and generalizing the framework to non-linear models
>
> On the theoretical front, characterizing the Hessian in arbitrary non-linear models remains challenging. In particular, the Hessian of the linear model admits a tractable form for which we can analytically relate its condition number to the singular values of the task-specific data matrix and the shared weight matrix. As a result, our theoretical guarantees on gradient updates with respect to the feature extractor $\theta$, which relies heavily on rigorous matrix analysis, are yet to be generalized to non-linear models. **The proposed regularizations, however, are applicable to bounding the condition number of general matrices, including the Hessian of non-linear models**.
>
> Hence, we have tested the empirical performance of the immunization framework on various non-linear models including ResNet18 and ViT with linear probing. As demonstrated in Sec. 5.2, the results validate the practical effectiveness.

---

### Decision · Program_Chairs · 2025-05-01

**Decision:**

Accept (oral)

**Comment:**

This paper presents a novel and theoretically grounded framework for model immunization through condition number manipulation. The authors propose differentiable regularizers that make fine-tuning on harmful tasks more difficult while preserving performance on benign tasks. The paper makes strong theoretical contributions with provable guarantees in linear models, and further demonstrates empirical effectiveness on both linear and deep nonlinear models (ResNet18, ViT, and CVAE).

Pros:
* the use of condition numbers to formalize and optimize for model immunization is principled and innovative.
* the theoretical analysis is rigorous and well-supported, particularly for linear models.
* the authors addressed all reviewer concerns during the rebuttal, including adding generative model results and showing fine-tuning behavior over epochs.
* experimental results show promising immunization effects without degrading primary task performance.
* reviewers appreciated the clarity of the writing, solid empirical support, and potential for broader impact in model safety and unlearning research.

Cons:
* the primary limitation is the theoretical framework's reliance on linear models, with extensions to deep models being empirical only.
* the evaluation metric (RIR) is task-appropriate but some reviewers suggested it should be supplemented with more direct measures of immunization in deep models.

Overall, the paper introduces a promising, technically sound approach to model immunization, with theoretical depth and convincing experiments. Despite limitations in generalizing theory to nonlinear models, the extensive rebuttal and new experiments strengthen its contribution. I recommend acceptance.